# Causality Analysis with Information Geometry: A Comparison

**DOI:** 10.3390/e25050806

**Published:** 2023-05-16

**Authors:** Heng Jie Choong, Eun-jin Kim, Fei He

**Affiliations:** 1Centre for Fluid and Complex Systems, Coventry University, Coventry CV1 5FB, UK; ejk92122@gmail.com; 2Centre for Computational Science and Mathematical Modelling, Coventry University, Coventry CV1 5FB, UK; fei.he@coventry.ac.uk

**Keywords:** causality, information geometry, Transfer Entropy, Granger Causality, information causal rate, signal processing, nonlinear models, non-stationary, probability distribution

## Abstract

The quantification of causality is vital for understanding various important phenomena in nature and laboratories, such as brain networks, environmental dynamics, and pathologies. The two most widely used methods for measuring causality are Granger Causality (GC) and Transfer Entropy (TE), which rely on measuring the improvement in the prediction of one process based on the knowledge of another process at an earlier time. However, they have their own limitations, e.g., in applications to nonlinear, non-stationary data, or non-parametric models. In this study, we propose an alternative approach to quantify causality through information geometry that overcomes such limitations. Specifically, based on the information rate that measures the rate of change of the time-dependent distribution, we develop a model-free approach called information rate causality that captures the occurrence of the causality based on the change in the distribution of one process caused by another. This measurement is suitable for analyzing numerically generated non-stationary, nonlinear data. The latter are generated by simulating different types of discrete autoregressive models which contain linear and nonlinear interactions in unidirectional and bidirectional time-series signals. Our results show that information rate causalitycan capture the coupling of both linear and nonlinear data better than GC and TE in the several examples explored in the paper.

## 1. Introduction

The study of causality aims to explore cause-and-effect relationships between processes. In general, causality can be investigated in two categories: causal discovery and causal analysis. Causal discovery shows the inherent causal relationship within the dataset by analyzing and creating causal models based on graph theory. The construction of these models can be achieved via algorithmic dynamics and evaluating the algorithms against the observed data [1,2,3]. Generally, the complexity of the models’ networks is characterized and measured using graph-theoretic measures. A recent approach using classical information theory, such as Shannon entropy, has been considered as an alternative measurement, but its high dependence on the distribution can lead to spurious disparate values for the same complexity, as shown in [4].

Alternatively, causal analysis, which is the focus of this paper, studies the potential changes in a system caused by another. In practical terms, causal discovery requires more detailed background information about the provided dataset to accurately conduct the study. However, the causal analysis does not require detailed underlying knowledge of the connectivity embedded within the dataset [3,5]. Among the various analysis methods used to quantify causality, Granger causality (GC) and Transfer Entropy (TE) are extensively used in various fields, including economics, climate, education, meteorology, neuroscience, and more [5,6,7,8,9,10,11,12,13,14,15,16]. For instance, Rebecca M. Pullon et al. utilized GC to demonstrate the decrease in cortical information flow throughout the brain as the subject loses responsiveness [10]; Yunyuan Gao et al. used TE to investigate the coupling strength of the brain between the motor and sensory areas [15]. Both of these methods are derived from the same notion of examining the reliability of a present process towards the past process(es). Despite sharing the same notion, their approaches to quantifying causality are completely different. GC evaluates the dependency based on signal models, which means modelling the signals incorrectly can lead to the wrong conclusion about the causality [11,17,18]. On the other hand, TE is model-free and employs the statistical dependency approach to access causality. However, it is subject to the estimation of the probability distribution, such as the number of bin size and the high dimensionality in the distribution due to the number of lags [19,20].

More specifically, inspired by Norbert Wiener, Clive Granger introduced GC [6,21], which suggested that the causality of two stochastic processes ($x_{1}\left(t\right)$ and $x_{2}\left(t\right)$) exists if and only if one of the processes is able to predict another. For instance, if including the past observation of $x_{1}\left(t\right)$ along with $x_{2}\left(t\right)$ improves the prediction of the current state of $x_{2}\left(t\right)$ compared to using only past information of $x_{2}\left(t\right)$, then this would suggest that $x_{1}\left(t\right)$ is Granger-causing $x_{2}\left(t\right)$. This predictivity is quantitatively studied by comparing the errors of the autoregressive model (linear regressive model) that is used in modelling the signals [6,18]. As natural time series such as electroencephalogram (EEG) often exhibit oscillatory aspects, it is interesting to study the spectrum of the GC. John Geweke has discussed the evaluation of GC in the frequency domain instead of the time domain [17,22,23]. Additionally, the time-varying frequency domain of the GC that is studied in [18,24,25] is able to show both the spectrum of the GC and the period of causation occurrence. The frequency domain of GC is the spectral of GC which shows the amplitude of GC correlated to a specific frequency of the entire signals, while the time-varying frequency of GC shows when the coupling of the signals occurs alongside the spectral of GC.

When utilizing the GC, it is necessary to model the provided signals using a linear autoregressive model [17,18]. However, this linear model cannot accommodate possible nonlinear and non-stationary effects within real-world signals, such as the brain EEG signals [26,27]. Therefore, instead of studying the causality through deterministic or parametric linear models, causality can be statistically quantified using information-theoretic measures like TE [19]. In the event of two stochastic processes, TE measures the additional information required from one process to achieve the current realization of another process [28]. Therefore, TE is not limited to any specific model or linearity assumption. Although TE uses a different approach in evaluating the causality, it can yield similar results to GC for a Gaussian process, as discussed analytically by Lionel Barnett et al. [29].

Alternatively, the causality can be investigated by examining the changes in the probability distribution that are caused by the inclusion of the other variables [30]. In a statistical space, the change in distribution is known as distance and can be quantified via Kullback–Leibler (KL) divergence. However, KL divergence cannot be well-defined as a distance, as it is not symmetrical and not path-dependent. Hence, information rate and information length are used with the intention to have the symmetrical and path-dependent measurement of the distance [30,31,32,33]. Information rate quantifies the rate of the temporal change in distributions, while information length measures the total statistical change in the process along the time [30,32]. Refs. [30,31] proposed an information-geometric causality measure, the so-called *information rate causality* based on the information rate, by quantifying causality through the effects of the information rate (see also [34]) and applied it to a solvable, linear Gaussian process (Kramer model) which has an exact time-dependent distribution.

In this paper, we will further develop this information rate causality for analyzing the numerically generated time-series signals that are in general time-varying and nonlinear and compare it with the GC and TE. Specifically, we will generate the signals by simulating the (discrete) autoregressive equations which contain unidirectional or bidirectional interactions. For the GC, both the frequency domain and time varying of GC will be presented. For the TE, the signals are evaluated as a whole at different lags. Next, the net TE is calculated by taking the difference in the TE from both directions. For instance, $T_{1,2}^{net}=T_{1\to 2}-T_{2\to 1}$ represents the net TE of signals 1 and 2. The net TE enables us to identify the appropriate lag(s) that yield significant results in TE. Using the knowledge of the appropriate lag(s) from the net TE, the TE is calculated through the window sliding approach, with each small window of the signals evaluating TE at the designated lag. Similarly, information rate causality will be evaluated through the window sliding approach with the information rate calculated within each window. In the scenario where the signals are coupled, it is expected that there will be spikes in the information rate causality, indicating changes in the distribution of one signal caused by another. Note that the methodology and concepts of the analyses used in this paper are mainly from the previous works [17,18,19,28,30,32].

The remainder of this paper is organized as follows. In Section 2, we briefly review the concepts for GC, TE, and information rate causality along with their implementation. The three different types of autoregressive models—unidirectional, interchange unidirectional, and bidirectional models—are introduced and analyzed in Section 3, Section 4 and Section 5 respectively. Section 6 contains our conclusions. Appendix B, Appendix C and Appendix D contain some background theory to make the paper self-contained. As a reference, the basic model of continuous coupling is studied and analyzed in Appendix E.

## 2. Methodologies

### 2.1. Granger Causality

The general idea behind GC is to evaluate the dependency of one process on another process [17,18]. This dependency is calculated by comparing the errors/noises of the modelled signals through the autoregressive model. Consider two stochastic processes, namely $x_{1}\left(t\right)$ and $x_{2}\left(t\right)$, which can be modelled using the information from their respective signals (as shown in Equations (1) and (2)) or including some information from each other (as shown in Equations (3) and (4)).
(1)$$\begin{array}{cc} x_{1}\left(t\right) & =\sum_{i=1}^{n}a_{i}x_{1}\left(t-i\right)+\epsilon_{1}\left(t\right), \end{array}$$
(2)$$\begin{array}{cc} x_{2}\left(t\right) & =\sum_{i=1}^{n}c_{i}x_{2}\left(t-i\right)+\epsilon_{2}\left(t\right), \end{array}$$
(3)$$\begin{array}{cc} x_{1}\left(t\right) & =\sum_{i=1}^{n}a_{i}^{*}x_{1}\left(t-i\right)+\sum_{i=1}^{n}b_{i}^{*}x_{2}\left(t-i\right)+\epsilon_{1}^{*}\left(t\right), \end{array}$$
(4)$$\begin{array}{cc} x_{2}\left(t\right) & =\sum_{i=1}^{n}c_{i}^{*}x_{1}\left(t-i\right)+\sum_{i=1}^{n}d_{i}^{*}x_{2}\left(t-i\right)+\epsilon_{2}^{*}\left(t\right). \end{array}$$

Here, $\{a_{i},c_{i},a_{i}^{*},b_{i}^{*},c_{i}^{*},d_{i}^{*}\}\in R$ are the constant parameters that represent the fractions of contribution from the past observations towards $x_{1}\left(t\right)$ and $x_{2}\left(t\right)$. $\epsilon_{i\in [1,2]}$ is the uncorrelated external additive noise needed for modelling the processes with variance $\Sigma_{ii,i\in [1,2]}$. $\epsilon_{i\in [1,2]}^{*}$ is the correlated noise with the variance $\Sigma_{ij,(i,j)\in [1,2]}^{*}$, which can be represented by the covariance matrix $\Sigma =\left[\begin{array}{cc} \Sigma_{11}^{*} & \Sigma_{12}^{*}\\ \Sigma_{21}^{*} & \Sigma_{22}^{*} \end{array}\right]$.

The total interdependence or total causality index ($F_{x_{1}\cdot x_{2}}$) between $x_{1}\left(t\right)$ and $x_{2}\left(t\right)$ is composed of two directional causalities ($F_{x_{1}\to x_{2}}$ and $F_{x_{2}\to x_{1}}$) and one instantaneous causality ($F_{x_{1}↔x_{2}}$). These are defined in Equations (5) to (8).
(5)$$\begin{array}{cc} F_{x_{1}\cdot x_{2}} & =\log\frac{\Sigma_{11}\Sigma_{22}}{\left|\Sigma \right|}=\log\frac{\Sigma_{11}\Sigma_{22}}{\Sigma_{11}^{*}\Sigma_{22}^{*}-{\left(\Sigma_{12}^{*}\right)}^{2}}=F_{x_{1}\to x_{2}}+F_{x_{2}\to x_{1}}+F_{x_{1}↔x_{2}}, \end{array}$$
(6)$$\begin{array}{cc} F_{x_{1}\to x_{2}} & =\log\frac{\Sigma_{11}}{\Sigma_{11}^{*}}, \end{array}$$
(7)$$\begin{array}{cc} F_{x_{2}\to x_{1}} & =\log\frac{\Sigma_{22}}{\Sigma_{22}^{*}}, \end{array}$$
(8)$$\begin{array}{cc} F_{x_{1}↔x_{2}} & =\log\frac{\Sigma_{11}^{*}\Sigma_{22}^{*}}{\left|\Sigma \right|}. \end{array}$$

In conjunction, the spectral of the GC can be evaluated in the frequency domain (ω=2πf,f≡ frequency) using the spectral matrix S(ω), transfer matrix H(ω), and the covariance matrix Σ. They are related according to Equation (9).
(9)$$\begin{array}{c} S\left(\omega \right)=\left[\begin{array}{cc} S_{11}\left(\omega \right) & S_{12}\left(\omega \right)\\ S_{21}\left(\omega \right) & S_{22}\left(\omega \right) \end{array}\right]=\langle x\left(\omega \right)x^{†}\left(\omega \right)\rangle =H\left(\omega \right)\Sigma H^{†}\left(\omega \right). \end{array}$$

Here, $x\left(\omega \right)=\left[\begin{array}{c} x_{1}\left(\omega \right)\\ x_{2}\left(\omega \right) \end{array}\right]$, $H\left(\omega \right)=\left[\begin{array}{cc} H_{11}\left(\omega \right) & H_{12}\left(\omega \right)\\ H_{21}\left(\omega \right) & H_{22}\left(\omega \right) \end{array}\right]$ = $\left[\begin{array}{cc} 1-\sum_{i=1}^{n}a_{i}^{*}e^{-j\omega i} & -\sum_{i=1}^{n}b_{i}^{*}e^{-j\omega i}\\ -\sum_{i=1}^{n}c_{i}^{*}e^{-j\omega i} & 1-\sum_{i=1}^{n}d_{i}^{*}e^{-j\omega i} \end{array}\right]$; $\langle \cdot \rangle$ is defined as an ensemble average and † is denoted as the complex conjugate transpose of a matrix.

Similar to the analysis in the time domain, the total independence in the frequency domain between $x_{1}\left(\omega \right)$ and $x_{2}\left(\omega \right)$ ($I_{x_{1}\cdot x_{2}}$) also consists of two directional causalities ($I_{x_{1}\to x_{2}}$ and $I_{x_{2}\to x_{1}}$) and one instantaneous causality ($I_{x_{1}↔x_{2}}$). They are expressed as Equations (10) to (13).
(10)$$\begin{array}{cc} I_{x_{1}\cdot x_{2}} & =\log\frac{S_{11}\left(\omega \right)S_{22}\left(\omega \right)}{\left|S(\omega )\right|}=I_{x_{1}\to x_{2}}+I_{x_{2}\to x_{1}}+I_{x_{1}↔x_{2}}. \end{array}$$
(11)$$\begin{array}{cc} I_{x_{1}\to x_{2}} & =\log\frac{S_{22}\left(\omega \right)}{{\hat{H}}_{22}\left(\omega \right)\Sigma_{22}{\bar{\hat{H}}}_{22}\left(\omega \right)}, \end{array}$$
(12)$$\begin{array}{cc} I_{x_{2}\to x_{1}} & =\log\frac{S_{11}\left(\omega \right)}{{\tilde{H}}_{11}\left(\omega \right)\Sigma_{11}{\bar{\tilde{H}}}_{11}\left(\omega \right)}, \end{array}$$
(13)$$\begin{array}{cc} I_{x_{1}↔x_{2}} & =\log\frac{\left({\tilde{H}}_{11}\left(\omega \right)\Sigma_{11}{\bar{\tilde{H}}}_{11}\left(\omega \right)\right)\left({\hat{H}}_{22}\left(\omega \right)\Sigma_{22}{\bar{\hat{H}}}_{22}\left(\omega \right)\right)}{\left|S(\omega )\right|}. \end{array}$$

Here, ${\hat{H}}_{22}\left(\omega \right)=H_{22}\left(\omega \right)+\frac{\Sigma_{12}^{*}}{\Sigma_{22}^{*}}H_{21}\left(\omega \right)$ and ${\tilde{H}}_{11}\left(\omega \right)=H_{11}\left(\omega \right)+\frac{\Sigma_{12}^{*}}{\Sigma_{11}^{*}}H_{12}\left(\omega \right)$; $\bar{\left[\cdot \right]}$ is denoted as the complex conjugate of the element.

In this paper, the frequency domain and the time-varying frequency domain of GC will be calculated through the non-parametric method. The general idea of this method is to decompose the spectral matrix S(ω) into the transfer matrix H(ω) and the covariance of the noises Σ. This decomposition can be achieved using Wilson’s algorithm [18,24,25,35,36]. The spectral matrices for the frequency domain and the time-varying frequency domain are expressed as Equations (14) and (15), respectively.
(14)$$\begin{array}{cc} S(\omega ) & =\frac{\langle x_{i}\left(\omega \right)\bar{x_{j}\left(\omega \right)}\rangle }{T},\;i,j\in R, \end{array}$$
(15)$$\begin{array}{cc} S(t,\omega ) & =\frac{\langle s_{i}\left(t,\omega \right)\bar{s_{j}\left(t,\omega \right)}\rangle }{T},\;i,j\in R. \end{array}$$

Here, *T* denotes the total period of the signal. $x{\left(\omega \right)}_{n|n\in [i,j]}$ is the discrete Fourier transform of the signal and $s_{n|n\in [i,j]}\left(t,\omega \right)$ is the short-time Fourier transform of the signal $x{\left(t\right)}_{n|n\in [i,j]}$. Note that $\bar{\left[\cdot \right]}$ denotes the complex conjugate. The Hann function window is used when evaluating the short-time Fourier transform. The window will move through the whole series with half of the data points being overlapped. The general flow of the procedure is shown in Figure 1a.

### 2.2. Transfer Entropy

TE is a model-free approach used to calculate causality by evaluating the dependency between processes. The general expression of TE is given by Equation (17), which measures the additional information required for the realization of one state of the process ($x_{1}\left(t\right)$) depending on the past states of the processes (e.g., $x_{1}\left(t-1\right)$ and $x_{2}\left(t-1\right)$) [19,28,37].
(16)$$\begin{array}{cc} TE_{x_{2}\left(t\right)\to x_{1}\left(t\right)} & =\sum_{x\in x_{1}\left(t\right),y\in x_{2}\left(t\right)}p\left(x_{n+1},x_{n}^{\left(k\right)},y_{n}^{\left(l\right)}\right)\log_{2}\frac{p(x_{n+1}|x_{n}^{\left(k\right)},y_{n}^{\left(l\right)})}{p(x_{n+1}|x_{n}^{\left(k\right)})} \end{array}$$
(17)$$\begin{array}{cc} \; & =\sum_{x\in x_{1}\left(t\right),y\in x_{2}\left(t\right)}p\left(x_{n+1},x_{n}^{\left(k\right)},y_{n}^{\left(l\right)}\right)\log_{2}\frac{p\left(x_{n+1},x_{n}^{\left(k\right)},y_{n}^{\left(l\right)}\right)p\left(x_{n}^{\left(k\right)}\right)}{p\left(x_{n+1},x_{n}^{\left(k\right)}\right)p\left(x_{n}^{\left(k\right)},y_{n}^{\left(l\right)}\right)}. \end{array}$$

Here, $x_{n+1}$ represents the state of $x_{1}\left(t\right)$ at the (n+1)-th moment, $x_{n}^{\left(k\right)}$ represents the state at *n*-th moment of $x_{1}\left(t\right)$ consisting of *k* previous states of $x_{1}\left(t\right)$, $y_{n}^{\left(l\right)}$ represents the state at *n*-th moment of $x_{2}\left(t\right)$ consisting of *l* previous states. Note that the previous states of *k* and *l* are arbitrary depending on the interest of the study. For instance, in a collective observed stochastic process $X=(x_{1},x_{2},\ldots ,x_{10})$, $x_{8}^{\left(2\right)}=\left(x_{n},x_{m}\right)\;|n,m\in \left[1,8\right]\to \left(x_{1},x_{2}\right)$ or $\left(x_{3},x_{8}\right)$, etc. Thus, TE quantifies the additional information needed for the realization of $x_{1}\left(t\right)$ at (n+1) from $x_{2}\left(t\right)$ with the assumption that $x_{1}\left(t\right)$ is independent of $x_{2}\left(t\right)$. If $x_{2}\left(t\right)$ has no impact on the future evolution of $x_{1}\left(t\right)$, and hence $p\left(x_{n+1}|x_{n}^{\left(k\right)},y_{n}^{\left(l\right)}\right)=p\left(x_{n+1}|x_{n}^{\left(k\right)}\right)$, then $T_{Y\to X}=0$.

For the TE calculation, the probability distribution p(…) will be estimated via histogram with the bin size of 5, as this bin size is determined by the cubic root of Rice’s rule to accommodate the 3-dimensional of joint probability within TE ($p(x_{n+1},x_{n}^{\left(1\right)},y_{n}^{\left(1\right)})$). Using a larger number of bin sizes will not be able to properly depict the distribution and it will bring a spurious result in calculating the TE. To simplify the analysis, the number of past values of *k* and *l* is set to 1 (k=l=1) in this paper. Two different evaluations will be conducted for TE. First, assuming that the processes are stationary, the net TE ($TE_{x_{1}\left(t\right),x_{2}\left(t\right)}=TE_{x_{1}\left(t\right)\to x_{2}\left(t\right)}-TE_{x_{2}\left(t\right)\to x_{1}\left(t\right)}$) is evaluated on the entire processes at different lags. This will enable us to choose the appropriate past value(s)/lag(s) in evaluating the TE based on the significance of the net TE. Second, using the knowledge of the appropriate lag from the net TE, the TE is next evaluated via a sliding window approach. This sliding window approach is used to calculate the TE at each instance of the interested window. In this sliding window approach, a small number of data points of the stochastic processes is sampled within the window and the TE is calculated. These calculations are sketched in Figure 1b.

### 2.3. Information Rate Causality

In a dimensionless statistical manifold, the distance between two probability distributions is defined by their statistical difference. One commonly used measure of this difference is the KL divergence/relative entropy, but this measure is not symmetrical and not path-dependent. For the time-dependent probability distribution, the changes in the distribution along with time can be measured through the information rate and information length. The information rate (Γ(t)) quantifies the rate of change of the distribution, while information length (L) measures the total change of the distribution. These measures are defined and expressed as Equations (18) and (19) [30,31,32].
(18)$$\begin{array}{cc} \Gamma^{2}\left(t\right) & =\int dx\;p\left(x,t\right){\left[\partial_{t}\ln p(x,t)\right]}^{2}=4\int dx\;{\left[\partial_{t}\sqrt{p(x,t)}\right]}^{2}, \end{array}$$
(19)$$\begin{array}{cc} L & =\int dt\;\Gamma (t). \end{array}$$

Here p(x,t) is the probability distribution of a stochastic process *x* at the time *t*.

For studying causality, we consider two stochastic processes ($x_{1}\left(t\right)$ and $x_{2}\left(t\right)$) and the information rate for a joint probability distribution is evaluated. The causality between these processes can be quantified by the changes in the information rate (statistical changes or the changes in distribution) while having another process remain static. For instance, the information rate of $x_{2}\left(t\right)$ causing $x_{1}\left(t\right)$ is defined as Equation (21) and called information rate causality in this paper.
(20)$$\begin{array}{cc} \Gamma^{2}{\left(t\right)}_{x_{2}\left(t\right)\to x_{1}\left(t\right)} & =\int dx_{1}\;dx_{2}\;p\left(x_{1}\left(t\right),x_{2}\left(t_{i}\right)\right){\left[\partial_{t}\ln p(x_{1}\left(t\right),x_{2}\left(t_{i}\right))\right]}^{2} \end{array}$$
(21)$$\begin{array}{cc} \; & =4\int dx_{1}\;dx_{2}\;{\left[\partial_{t}\sqrt{p(x_{1}\left(t\right),x_{2}\left(t_{i}\right))}\right]}^{2}. \end{array}$$

Referring to Figure 1c, here $t_{i}$ is denoted as the reference time that remains static in the process. Therefore, $p(x_{1}\left(t\right),x_{2}\left(t_{i}\right))$ is a probability distribution that is sampled by having the process $x_{2}\left(t\right)$ remain static at time $t_{i}$ while the process $x_{1}\left(t\right)$ evolves along the time *t*. Since information rate can only quantify the changes in the distribution, therefore the *information rate causality* is evaluated within each window of interest instead of the whole process of the time series to determine whether the coupling of the processes is still persists. To accommodate the calculation, a discretized version of Equation (21) is used and it is expressed as Equation (22).
(22)$$\begin{array}{c} \Gamma^{2}{\left(t\right)}_{x_{2}\to x_{1}}=4\sum_{x_{1}\in X_{1},x_{2}\in X_{2}}\frac{\Delta x_{1}\Delta x_{2}}{{\left[\Delta t\right]}^{2}}{\left[\sqrt{p(x_{1}\left(t+\Delta t\right),x_{2}\left(t_{i}\right))}-\sqrt{p(x_{1}\left(t\right),x_{2}\left(t_{i}\right))}\right]}^{2} \end{array}$$

For calculating the information rate, the joint probability distribution for two processes is estimated via the histogram with the bin size (*b*) determined by the square root of Rice’s rule, as expressed in Equation (23). Rice’s rule is used because it is able to appropriately determines the bin size to sample the obtained data [38,39] for 1-dimensional distribution. Hence, the square root of Rice’s rule is to accommodate the 2-dimensional distribution in this case.
(23)$$\begin{array}{c} b=\sqrt{\left\lceil 2\sqrt[{3}]{n}\right\rceil },\;n=\mathrm{number}\;\mathrm{of}\;\mathrm{sampled}\;\mathrm{data}. \end{array}$$

In this paper, we evaluate the information rate causality by employing Equation (22) to examine the impact of one process, which remains static in time, on the distribution of another process within a specific time window. To estimate the joint probability distribution utilized in Equation (22), a histogram is employed with a bin size determined by Equation (23). Figure 1c illustrates the overall procedure of this evaluation.

### 2.4. Summary of the Procedure: Data and Analysis

We further develop *information rate causality* (refer to Section 2.3) for the analysis of our numerically generated data and compare it with the non-parametric GC (refer to Section 2.1) and window sliding TE (refer to Section 2.2). Our numerical data are generated by simulating different discrete autoregressive models covering both linear and nonlinear models. The simulation is conducted for 10,000 trials with each trial running for 25 s (physical time) at 200 Hz (physical sampling frequency). Different coupling/causal relationships between the signals are considered in this paper, such as unidirectional (refer to Section 3), interchanging unidirectional (refer to Section 4), and bidirectional (refer to Section 5). Note that the noncoupling cases are also considered in this paper to check the robustness of the causality analyses; specifically, we refer to the uncoupling of $x_{1}\left(t\right)$ towards $x_{2}\left(t\right)$ after the physical time 10 s. Using simulated signals, the causality analyses (GC, TE, *information rate causaltiy*) are conducted according to the sketch shown in Figure 1.

To be consistent with the data points used for each window, each sampling window will contain 0.5 s of data points with an overlap of 0.25 s (0.5×0.5s), which is equivalent to 100 data points for one simulation. Since the models are simulated for 10,000 trials, each window will consist of $1\times 10^{6}$ sample points (100points×10,000trials). Note that this window is not applied to the frequency domain of GC and the net TE as both are calculated based on the whole series.

### 2.5. Summary of the Different Models and Key Results

Prior to the discussion of different models in Section 3, Section 4 and Section 5 with the implementation of different analysis methods, we provide the key results/findings in Table 1.

## 3. Unidirectional Model

In this part, unidirectional causality, where the first process ($x_{1}\left(t\right)$) influencing the second process ($x_{2}\left(t\right)$), is considered for both linear and nonlinear autoregressive models given by Equations (24) and (25), respectively. The causality between the processes occurs as the processes are coupled with each other through the Heaviside step function (H(…)).


**Linear model:**

(24)
$$\begin{array}{cc} x_{1}\left(t\right) & =0.55x_{1}\left(t-1\right)+\epsilon_{1}\left(t\right),\\ x_{2}\left(t\right) & =H\left(\tau -10\right)x_{1}\left(t-2\right)+\epsilon_{2}\left(t\right). \end{array}$$

**Nonlinear model:**

(25)
$$\begin{array}{cc} x_{1}\left(t\right) & =0.55x_{1}\left(t-1\right)e^{1-\left|x_{1}\left(t-1\right)\right|}+\epsilon_{1}\left(t\right),\\ x_{2}\left(t\right) & =H\left(\tau -10\right)x_{1}\left(t-2\right)+\epsilon_{2}\left(t\right). \end{array}$$



Here, *t* is the time steps and τ is the physical time. Both *t* and τ are related as τ=[physicalfrequency]×t. H(τ−10) is a Heaviside step function that allows the coupling to occur starting at the physical time of 10 s. $\epsilon_{n|n\in [1,2]}$ is the Gaussian noise that perturbates the systems. In this study, the noises are set to have zero mean ($\langle \epsilon_{n}\rangle =0|n\in \left[1,2\right]$) along with specific covariance/variance ($\Sigma_{nm}=\langle \left(\epsilon_{n}-\langle \epsilon_{n}\rangle \right)\left(\epsilon_{m}-\langle \epsilon_{m}\rangle \right)\rangle |n,m\in \left[1,2\right]$), which will be expressed in covariance matrix ($\left[\begin{array}{cc} \Sigma_{11} & \Sigma_{12}\\ \Sigma_{21} & \Sigma_{22} \end{array}\right]$). Due to the exponential term in the nonlinear model, the process will have non-stationary oscillation when the power of the exponential term becomes positive ($e^{R}\to \mathrm{diverge}$). Hence, two different values of noise covariance will be considered to study the stationary and non-stationary oscillation of the processes. They are labelled as large noise and small noise with the respective matrices expressed as $\left[\begin{array}{cc} \Sigma_{11} & \Sigma_{12}\\ \Sigma_{21} & \Sigma_{22} \end{array}\right]=\left[\begin{array}{cc} 1.0 & 0.0\\ 0.0 & 1.0 \end{array}\right]$ and $\left[\begin{array}{cc} \Sigma_{11} & \Sigma_{12}\\ \Sigma_{21} & \Sigma_{22} \end{array}\right]=\left[\begin{array}{cc} 0.005 & 0.0\\ 0.0 & 0.005 \end{array}\right]$. Note that the cross-covariance of the noises is set to zero ($\Sigma_{12}=\Sigma_{21}=0.0$) to ensure the coupling is purely due to the intrinsic interaction in the model but not through the mutual noises. The linear model will have stationary oscillation for either large or small noises, and hence only one noise will be used to simulate the linear model and large noise is chosen. The general structure of the models is shown in Figure 2.

To explore all possible combinations of cases, we investigate the coupling and noncoupling cases for both linear and nonlinear models. To this end, we investigate the following six difference cases: [linear: couple], [linear: noncouple], [nonlinear: large noise, coupling], [nonlinear: large noise, noncoupling], [nonlinear: small noise, coupling], and [nonlinear: small noise, noncoupling].

The result of the simulation for the Equations (24) and (25) is shown in Figure 3. Notice that the signals for the nonlinear model exhibit non-stationary oscillation when the perturbating noise is small, as shown in Figure 3e,f. This is due to the divergence of the positive power of the exponential term of Equation (25) ($e^{1-\left|x_{1}\left(t-1\right)\right|}=e^{R}\to \mathrm{diverge}\;|\epsilon_{1}\left(t\right)\equiv \mathrm{small}\;\mathrm{noise}$) and the influences of the previous state towards the current state ($x_{1}\left(t\right)\leftarrow 0.55x_{1}\left(t-1\right)$). The rest of the simulated signals have stationary oscillation, as shown in the phase space where the observed states are localized around $\left(x_{1},x_{2}\right)=\left(0,0\right)$.

### 3.1. Granger Causality

As shown in Figure 4, the coupling of $x_{1}$ to $x_{2}$ can be captured well through nonparametric GC analysis for stationary linear signals, as shown in Figure 4a. Concurrently, the detection of the causality works well for stationary nonlinear signals, as shown in Figure 4c. This is because the nonlinear exponential term is well-approximated to a constant value as the spectral matrix decomposed to the transfer matrix and covariance matrix of noise via Wilson’s algorithm. Note that the amplitude of the GC decreases for this nonlinear stationary signal. This is due to the contribution of the exponential term ($e^{-R}$) that eventually affects the $S_{22}\left(\omega \right)$ element of the spectral matrix (refer to Equation (11)) and hence the amplitude of the causality ($I_{x_{1}\to x_{2}}$). For non-stationary signals, the frequency domain of GC is still able to show the causality within the signals, but the time-varying frequency of GC cannot represent well the period of causation, as shown in Figure 4e. From the figure, we can see that the nonparametric GC gives a false result, as it suggests that the coupling between the processes occurs throughout the time. Similarly, the nonparametric GC analysis works fine in evaluating the non-causality cases for stationary cases but not so well for the non-stationary case (refer to Figure 4b,d,f. GC analysis fails to work on the signals oscillating non-stationary, as the signals cannot be modelled well via the linear autoregressive equations (refer to Equations (3) and (4)).

### 3.2. Transfer Entropy

The results from TE analysis are shown in Figure 5. By first assuming that the signals are stationary, the net TE ($TE_{X,Y}=TE_{X\to Y}-TE_{Y\to X}$) is first calculated to find the suitable lag for evaluating the sliding window TE later. Figure 5a,c,e show that the net TE between the signals is significant at the lag of 1. This is because the simulated models (Equations (24) and (25)) have the coupling occurring at one lag (compares $x_{1}\left(t\right):=x_{1}\left(t-1\right)$ and $x_{2}\left(t\right):=x_{1}\left(t-2\right)$). Next, the window sliding TE is conducted through the conditional probability (multidimensional probability) that contains the data from one previous lag (refer to Equation (17)) at each window of the evaluation of TE. As shown in Figure 5a,c,e, the window sliding TE is able to show the time (10 s physical time) when the causality occurs. Similarly, the TE is also able to capture the non-causality situation well for all the cases, as shown in Figure 5b,d,f. Even though TE is able to detect the causality between the signals for either linear stationary or nonlinear non-stationary cases, it requires extra inputs/assumptions (such as bin sizes, number of lags, and the dimension of the multidimensional probability) to work properly/accurately. The failure of TE in capturing the causality for the linear model (Equation (24)) is shown in Figure 6 when the lag is set to 9 instead of 1 when evaluating the TE.

### 3.3. Information Rate Causality

Both GC and TE detect causality by examining the increase in the predictability of the stochastic processes (measured by the inverse of entropy) when the past information is included. In comparison, information rate causality evaluates causality by quantifying the rate of change of the probability distribution of one variable conditioned on another, as noted previously. The results are shown in Figure 7 and Figure 8 where the changes of the distribution due to the causality between the processes are well-reflected via the changes in the information rate (Γ). For the cases of noncoupling, the information rate is not changing and remains constant, which would suggest that none of the signals cause the changes to the joint distribution (refer to Figure 7b,d,f). Hence, no causality occurs between the signals. Similarly, prior to the coupling of the signals, the causality is not seen, and hence the information rate again remains constant. Notice that the information rate does not stay at zero when no causality happens, it is due to the noises/spikes of the estimated distribution, as shown in Appendix A.

In the presence of the couplings, the changes in the distribution due to the causality among the signals can be observed via the changes in the information rate. For instance, the changes in the information rate due to the causality of $x_{1}\left(t\right)$ to $x_{2}\left(t\right)$ can be observed in Figure 7a,c,e (zoom-in: Figure 8a,c,e). Referring to the models, the signals are coupled with the lag difference of one ($x_{1}\left(t\right)=x_{1}\left(t-1\right)$ and $x_{2}\left(t\right)=x_{1}\left(t-2\right)$), and hence the peak of the information rate is observed after one lag. From Figure 8a,c,e, the changes in the information rate prior to the coupling (at physical time 10 s) are observed as the evaluation window of [9.75 s to 10.25 s] picks up the causality of the signals from [10 s to 10.25 s]. Referring to the starting time (near 0 s) of Figure 7e,f, there is the presence of a sharp difference in information rate. This is due to the non-stationary oscillation of $x_{1}\left(t\right)$ that causes the high disturbance of the distribution.

To demonstrate the capability of information rate causality to detect lag differences within a signal, we simulated and analyzed Equation (26), which has the coupling starting at 10 s with a lag difference of 9 (compares $x_{1}=x_{1}\left(t-1\right)$ and $x_{2}=x_{1}\left(t-10\right)$). Figure 9 shows the section of the information rate from 9.6 s to 10.4 s. The figure reveals that the peak of the information rate appears at the 9th time step (0.045 s) at every window of evaluation, which accurately reflects the lag difference specified in Equation (26). This result suggests that information rate causality can effectively capture the underlying causality of the signals at each evaluation window.
(26)$$\begin{array}{cc} x_{1}\left(t\right) & =0.55x_{1}\left(t-1\right)+\epsilon_{1}\left(t\right),\\ x_{2}\left(t\right) & =H\left(\tau -10\right)x_{1}\left(t-10\right)+\epsilon_{2}\left(t\right). \end{array}$$

## 4. Interchange of Unidirectional Model

In this part, similar models from Section 3 are used with slight modifications to study the interchange of the unidirectional causality. The models are modified to Equations (Equation 27) and (Equation 28) for linear and nonlinear models, respectively. The modification with the Heaviside step functions (H(…)) in the models portrays that the signal $x_{2}\left(t\right)$ causing $x_{1}\left(t\right)$ occurs prior to physical time (τ) 10 s, while the signal $x_{1}\left(t\right)$ causes $x_{2}\left(t\right)$ after 10 s. The general structure of Equations (27) and (28) is illustrated in Figure 10.


**Linear model:**

(27)
$$\begin{array}{cc} x_{1}\left(t\right) & =H\left(10-\tau \right)0.55x_{2}\left(t-1\right)+\epsilon_{1}\left(t\right),\\ x_{2}\left(t\right) & =H\left(\tau -10\right)x_{1}\left(t-2\right)+\epsilon_{2}\left(t\right). \end{array}$$

**Nonlinear model:**

(28)
$$\begin{array}{cc} x_{1}\left(t\right) & =H\left(10-\tau \right)0.55x_{2}\left(t-1\right)e^{1-\left|x_{2}\left(t-1\right)\right|}+\epsilon_{1}\left(t\right),\\ x_{2}\left(t\right) & =H\left(\tau -10\right)x_{1}\left(t-2\right)+\epsilon_{2}\left(t\right). \end{array}$$



Here *t* is the time-step and τ is the physical time. The Heaviside step functions are used with H(10−τ) suggesting the coupling occurs from the beginning of the simulation till the physical time 10 s conversely H(τ−10), suggesting the coupling occurs after 10 s. Similar to Section 3, 6 possible cases are simulated based on Equations (27) and (28). The simulated signals will be evaluated by GC, TE, and information rate causality.

The result of the simulations is shown in Figure 11, which suggests that all the signals oscillate stationary. Unlike Equation (25), in which the signal $x_{1}\left(t\right)$ is perturbated by its previous state by a factor of 0.55 ($0.55x_{1}\left(t-1\right)$) and the small noise ($\epsilon_{1}\left(t\right)$), Equation (28) is perturbated by the past state of $x_{2}\left(t\right)$ ($x_{2}\left(t-2\right)$) instead. Hence, the power of the exponential will always be negative; consequently, all the simulated signals have stationary oscillations. Note that the noncoupling is referring to the decoupling of $x_{1}\left(t\right)$ to $x_{2}\left(t\right)$ (H(τ−10)=0).

### 4.1. Granger Causality

Based on the models, the signal $x_{2}\left(t\right)$ will first cause $x_{1}\left(t\right)$ for the beginning of the process until the physical time of 10 s, and $x_{1}\left(t\right)$ causes $x_{2}\left(t\right)$ after the 10 s. This pattern is captured in the time-varying frequency of GC, as shown in Figure 12 for all the cases. Prior to 10 s, the $x_{1}\left(t\right)$ is caused by $x_{2}\left(t\right)$ by a factor of 0.55, and hence a similar factor of GC is shown in the time-varying frequency domain. Similarly, the coupling between the signals can also be observed in the frequency domain of GC. Notice that the [nonlinear: small noise] cases are able to show the significance of $x_{2}\left(t\right)$ causing $x_{1}\left(t\right)$ prior to 10 s because the small noise and the exponential term had $x_{2}\left(t\right)$ causing $x_{1}\left(t\right)$ increased to about 1.5 ($e^{1}\times 0.55\approx 1.5$).

### 4.2. Transfer Entropy

TE is able to show the correction direction of the causality between the signals if the correct lag is chosen. Therefore, having models (like Equations (27) and (28)) with the coupling occurring at different time lags cannot properly show the direction of the causality via the window sliding TE. For instance, the window sliding TE shows that $x_{2}\left(t\right)$ is causing the $x_{1}\left(t\right)$ when lag 0 is chosen, while the reverse is shown when lag 1 is used, as shown in Figure 13 (for lag 0) and Figure 14 (for lag 1). This is depicted well in the underlining models (Equations (27) and (28)), in which $x_{1}\left(t\right)$ causes $x_{2}\left(t\right)$ one lag later. As a result, TE cannot show the whole causality of the signals by just one lag.

### 4.3. Information Rate Causality

Alternatively, the information rate causality is able to show the coupling of the signals along with the lag of coupling in each window of the evaluation. As discussed, the information rate causality evaluates the causality by observing the changes in the probability distribution of one signal after it is influenced by another signal. As shown in Figure 15 and Figure 16, the information rate causality can captures the interchange of the coupling among the signals of the simulated models (Equations (27) and (28)). As discussed in TE, signal $x_{2}\left(t\right)$ is causing $x_{1}\left(t\right)$ at a lag of 0, and $x_{1}\left(t\right)$ causes $x_{2}\left(t\right)$ at a lag of 1. These lags can be observed at each window of the information rate evaluation. For instance, as shown in Figure 16, the information rate of 2 to 1 ($\Gamma^{2}$ 2 to 1) has a spike at the beginning of the time, while the information rate of 1 to 2 ($\Gamma^{2}$ 1 to 2) has a spike after one time step (one lag). Hence, the information rate causality is capable of showing the underlying causality of the signals.

## 5. Bidirectional Model

In this section, the bidirectional causality is studied by modifying the linear and nonlinear models from Section 3 as Equations (29) and (30), respectively. These modified models have the signals $x_{2}\left(t\right)$ continuously influencing $x_{1}\left(t\right)$ at all times t≥1. In addition, at time 10 s, the Heaviside step function (H(τ−10)) enables the coupling of $x_{1}\left(t\right)$ and $x_{2}\left(t\right)$, which allows $x_{1}\left(t\right)$ to influence $x_{2}\left(t\right)$ while the previous coupling remains intact. The general structure of the Equations (29) and (30) is illustrated in Figure 17.


**Linear model:**

(29)
$$\begin{array}{cc} x_{1}\left(t\right) & =0.55x_{2}\left(t-1\right)+\epsilon_{1}\left(t\right),\\ x_{2}\left(t\right) & =H\left(\tau -10\right)x_{1}\left(t-2\right)+\epsilon_{2}\left(t\right). \end{array}$$

**Nonlinear model:**

(30)
$$\begin{array}{cc} x_{1}\left(t\right) & =0.55x_{2}\left(t-1\right)e^{1-\left|x_{2}\left(t-1\right)\right|}+\epsilon_{1}\left(t\right),\\ x_{2}\left(t\right) & =H\left(\tau -10\right)x_{1}\left(t-2\right)+\epsilon_{2}\left(t\right). \end{array}$$



Here, *t* is the time step and τ is the physical time; H(τ−10) is the Heaviside step function that allows the coupling of the signal starting at a physical time of 10 s. Similar to the previous model, 6 cases are simulated and the signals are used to evaluate through GC, TE, and information rate causality. Recall that the noncoupling cases here are referring to the decoupling of $x_{1}\left(t\right)$ to $x_{2}\left(t\right)$ (H(τ−10)=0).

The simulation result from all aforementioned cases is shown in Figure 18. From the observation of the individual signal and the phase space, the scenario of [nonlinear: small noise, couple] has non-stationary oscillation after the coupling, as shown in Figure 18e. This is due to the divergence caused by the nonlinear exponential term embedded within the simulated equation ($e^{1-\left|x_{2}\left(t-1\right)\right|}=e^{R}$) along with the factor of 0.55 of the previous state of $x_{2}\left(t\right)$ ($0.55x_{2}\left(t-1\right)\ldots$). On the other hand, [nonlinear: small noise, noncouple] has a stationary oscillation, as shown in Figure 18f; this is because $x_{1}\left(t\right)$ only receives the input from $x_{2}\left(t\right)$, which keeps the power of the exponential term negative and hence results in a stationary oscillation.

### 5.1. Granger Causality

From the GC analysis, the coupling from $x_{2}\left(t\right)$ to $x_{1}\left(t\right)$ cannot be captured well in it except for the [nonlinear: small noise] cases. This result is due to a similar reason as discussed in Section 4, i.e., the coupling between the signals is affected by a factor of 1.5 ($e^{1}\times 0.55\approx 1.5$). Therefore, the rest of the cases cannot show the significant contrast of the causality from [$x_{2}\left(t\right)$ to $x_{1}\left(t\right)$] as shown in Figure 19. Furthermore, referring to the time-varying GC, the bidirectional coupling/causality between $x_{1}\left(t\right)$ and $x_{2}\left(t\right)$ that occurs after 10 s are not able to capture all the coupling cases.

### 5.2. Transfer Entropy

The results of TE shown in Figure 20 and Figure 21 once again are quite similar to those shown in Section 4. The window sliding TE, evaluated with lag 0, shows that signal $x_{2}\left(t\right)$ causes $x_{1}\left(t\right)$ throughout the time, as shown in Figure 20. For the coupling cases, the amplitude of the TE changes when $x_{1}\left(t\right)$ feedbacks to $x_{2}\left(t\right)$ (after time 10 s). Similar to Section 4, the window sliding TE only shows that signal $x_{1}\left(t\right)$ is causing $x_{2}\left(t\right)$ when evaluated at lag 1 for the coupling cases shown in Figure 21. Hence, TE cannot fully capture the causality between the signals with just one lag.

### 5.3. Information Rate Causality

The information rate causality analysis evaluates the causality through the change in probability distribution of an original signal after including another signal assuming it causes the original signal. The information rate causality is able to show the coupling among the signals well, as shown in Figure 22 and Figure 23. Similar to Section 4, prior to time 10 s, the causality from signal $x_{2}\left(t\right)$ to $x_{1}\left(t\right)$ can be depicted with the presence of the peak of $\Gamma^{2}$ 2 to 1. The peak presented at the beginning of the evaluation window suggests that the coupling occurs at lag 0. After 10 s, the mutual feedback between the signals is shown within the evaluation window of the information rate causality. This mutual causality between the signals is observed through the alternating oscillations of the peak occurrence in the information rate within the window. Hence, this can reflect well the underlying model of the signals. Note that this is different than Section 4, which is an interchange of unidirectional causality that has the feedback solely from one signal to the other without retaining information from itself.

## 6. Conclusions

In this paper, we proposed an alternative causal analysis method to the popular GC and TE for causality quantification. Specifically, based on information rate, which measures the rate of temporal change in the time-dependent distribution, we developed a model-free, information-geometric measure of causality—information rate causality—that is suitable for analyzing numerically generated non-stationary, nonlinear data. To compare the GC, TE, and information rate causality, we applied the methods to numerical data generated by simulating different types of discrete autoregressive models which contain linear and nonlinear interactions. In Section 3, we showed that information rate causality performs equally well compared to GC in the standard linear stationary signals. Later, we showed that the GC did not perform well for non-stationary and/or nonlinear signals. Furthermore, it failed to capture mutual feedback between the signals, as shown specifically in Section 5 in all the coupling cases. TE performed slightly better than GC, since it is model-independent. However, the information on the lag is needed to properly evaluate the TE, as shown in Section 4 and Section 5. In comparison with GC and TE, information rate causality has shown to be able to uncover the underlying mechanism of causality in the signals, such as the interchanging oscillatory feedback between the signals that is discussed in Section 5, which was not captured by either GC or TE.

While our results in this paper were obtained based on the interaction of two time series of different types of (non)linearity and (non)stationary, they have at least pointed out some limitations of GC and TE that can be resolved by employing information rate causality. It remains as future work to extend this work and to investigate a larger class of time series data, including real data such as EEG signals.

## Figures and Tables

**Figure 1 entropy-25-00806-f001:**
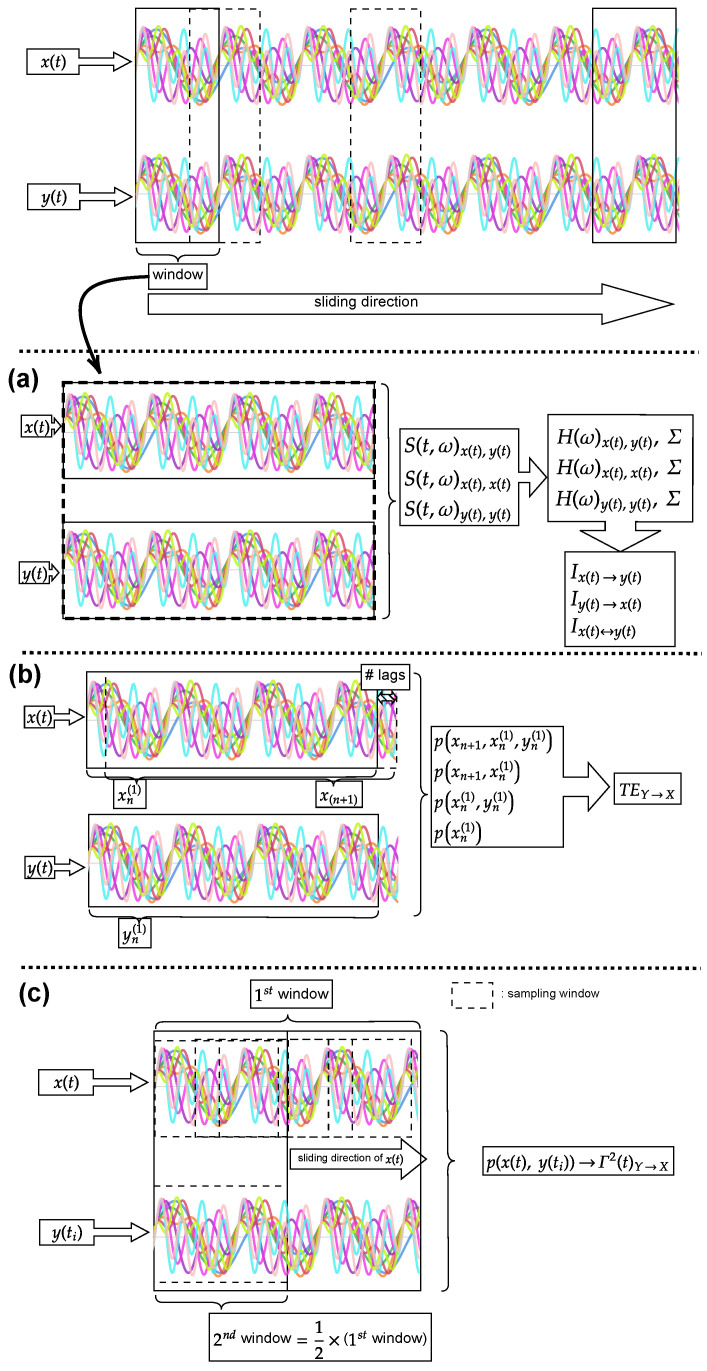
The procedure of implementing the causality analyses used in this paper, each colour of the lines in the image represents a different simulation. In this study, each window contains 0.5 s of data points and overlaps with the previous window by 0.25 s. The causality analyses are conducted within each window. (**a**) illustrates the components (S(…), H(…)) calculated within the windows for the non-parametric GC analysis. Refer to Section 2.1 to know the corresponding components. (**b**) shows the estimation of distributions p(…) for TE evaluation. The distributions p(…) are estimated based on the samples $x_{n}^{\left(1\right)}$, $x_{n+1}$, and $y_{n}^{\left(1\right)}$. Refer to Section 2.2 for the definition of TE. (**c**) demonstrates the evaluation of information rate causality. Each window (labelled as 1st window) is divided into two windows with the first half labelled as 2nd window. The distribution $p(x\left(t\right),y\left(t_{i}\right))$ estimates the evolution of distribution x(t) while $y(t_{i})$ is fixed at the 2nd window. Refer to Section 2.3 for the definition of information rate causality.

**Figure 2 entropy-25-00806-f002:**
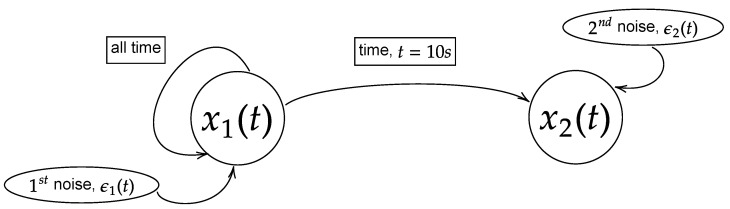
Model of the flow of information of the processes $x_{1}\left(t\right)$ and $x_{2}\left(t\right)$ for Equations (24) and (25). In this paper, the equations are simulated with the physical time of 25 s and sampling frequency of 200 Hz (5000 realizations) with either large noise or small noise. The coupling between the processes occurs at physical time of 10 s.

**Figure 3 entropy-25-00806-f003:**
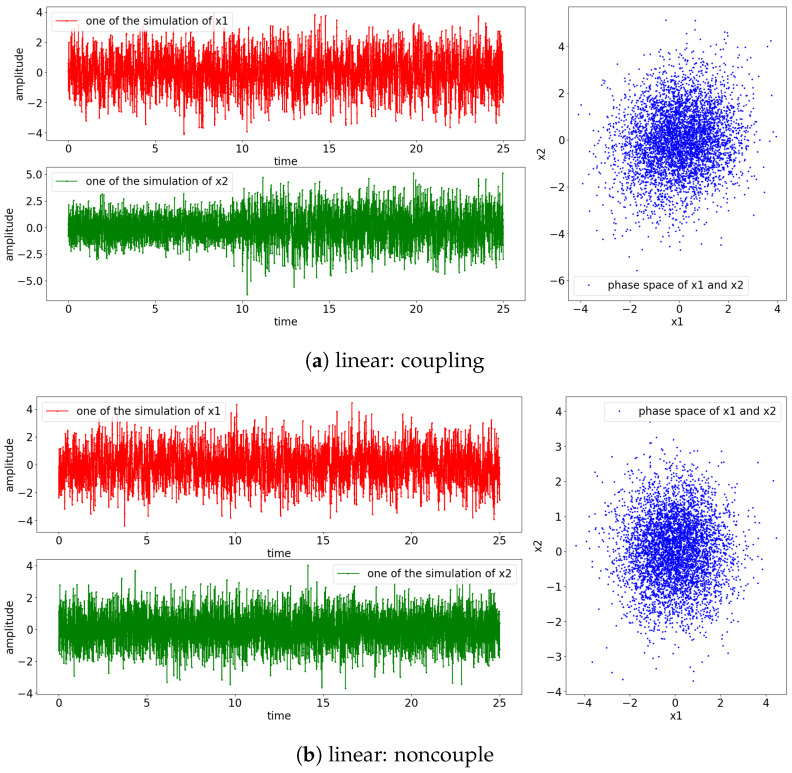
Result of the simulation based on Equations (24) and (25) (refer to Figure 2 for graphical explanation of the process). The [top left, red] is the result of $x_{1}$ and [bottom, green] is the result of $x_{2}$; [right blue] is the phase space of $x_{1}$ and $x_{2}$. Note that the coupling of the processes occurs at 10 s.

**Figure 4 entropy-25-00806-f004:**
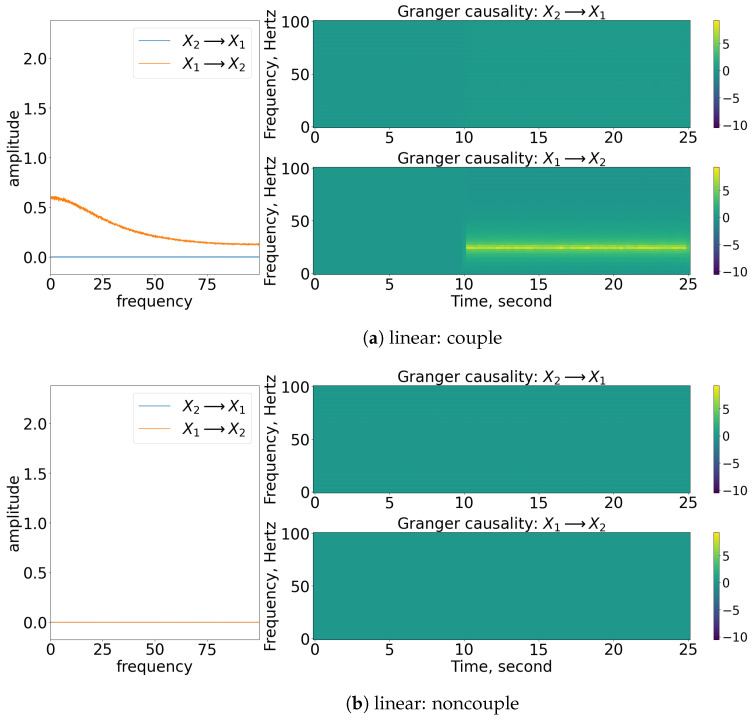
Result of the spectral and time-varying frequency of GC. Note that for each subfigure, the [left] shows the spectral (frequency domain) of GC, [blue] indicates $x_{2}$ causes $x_{1}$, while [orange] indicates $x_{1}$ causes $x_{2}$; [top, right] shows the time-varying frequency of GC of $x_{2}$ to $x_{1}$; [bottom, right] shows the time-varying frequency of GC of $x_{1}$ to $x_{2}$. The results shown are based on the processes in Equations (24) and (25). Refer to Figure 2 for the graphical explanation of the process.

**Figure 5 entropy-25-00806-f005:**
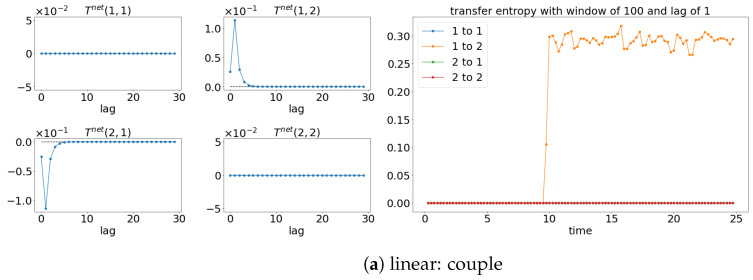
Result of the net TE and window sliding TE. Note that for each subfigure, [left, 4×4] depicts the net TE for the whole signals; while the [right] shows the window sliding TE. The results shown are based on the processes in Equations (24) and (25). Refer to Figure 2 for the graphical explanation of the process.

**Figure 6 entropy-25-00806-f006:**
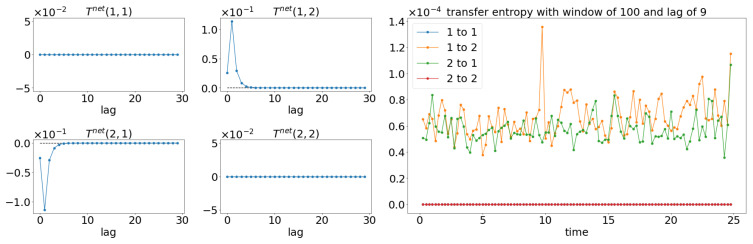
Failure of TE in capturing the causality between the signals $x_{1}$ and $x_{2}$ for [linear: couple] (refer to Equation (24)) when lag 9 is used for evaluation, as compared to Figure 5a, where the TE accurately shows that the coupling occurs after 10 s.

**Figure 7 entropy-25-00806-f007:**
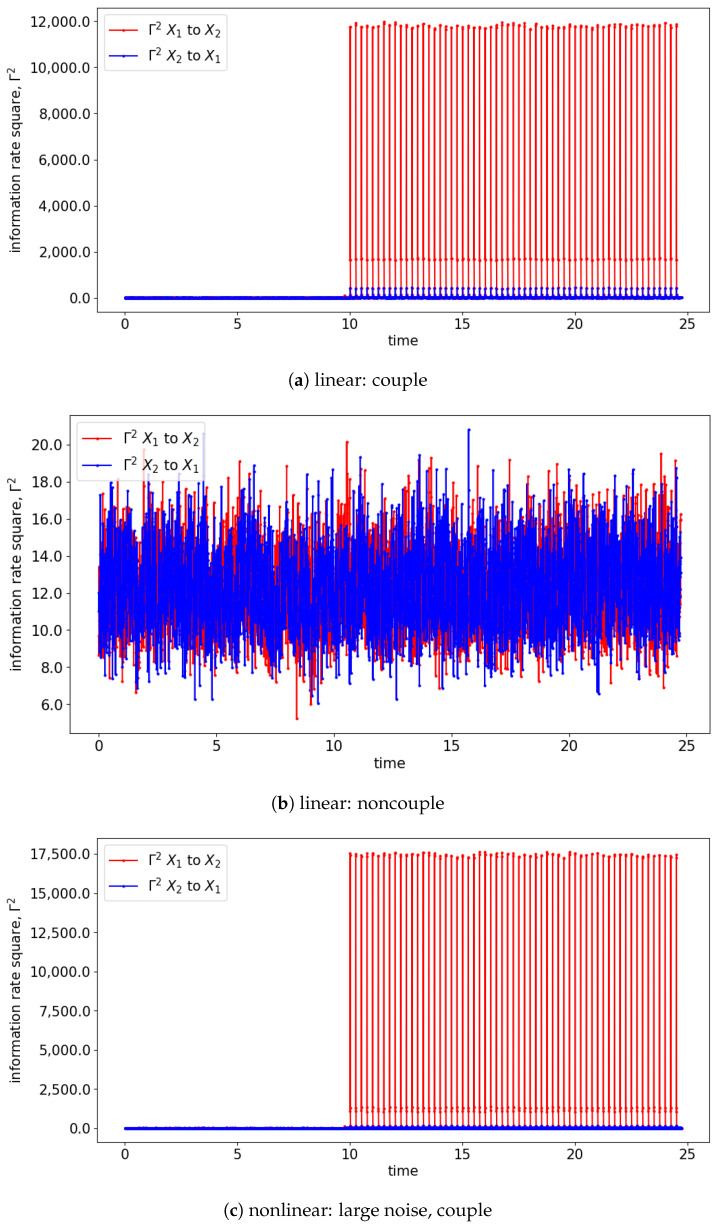
Result of information rate causality for Equations (24) and (25).

**Figure 8 entropy-25-00806-f008:**
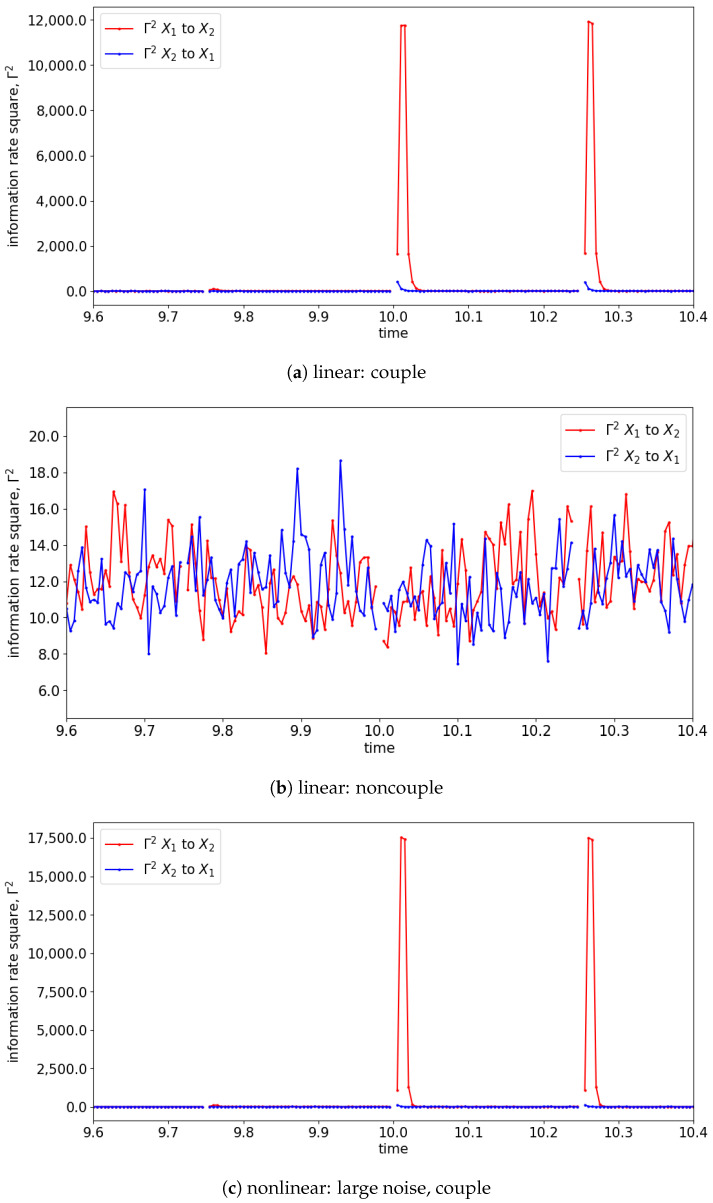
Zoom-in of Figure 7 for the information rate causality between 9.6 s and 10.4 s. Note that the change in the causality occurs at 10 s.

**Figure 9 entropy-25-00806-f009:**
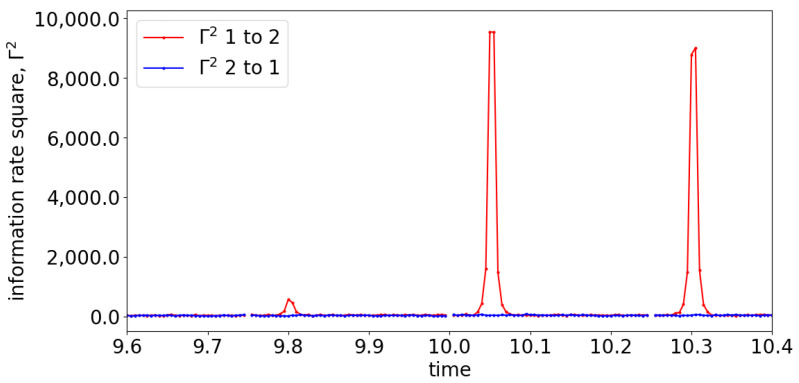
The information rate causality of Equation (26) between 9.6 s and 10.4 s where the causality occurs at 10 s.

**Figure 10 entropy-25-00806-f010:**
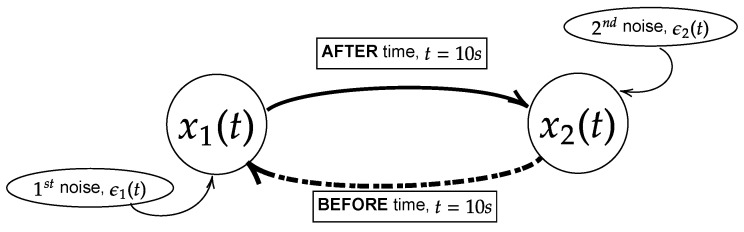
Model of the flow of information of the processes $x_{1}\left(t\right)$ and $x_{2}\left(t\right)$ for Equations (27) and (28). In this paper, the equations are simulated with the physical time of 25 s and sampling frequency of 200 Hertz (5000 realizations) with either large noise or small noise. Note that the process $x_{2}\left(t\right)$ coupling with $x_{1}\left(t\right)$ occurs before 10 s and the interchange of coupling occurs after 10 s. In the context of noncoupling, it is referring to H(τ−10)=0.

**Figure 11 entropy-25-00806-f011:**
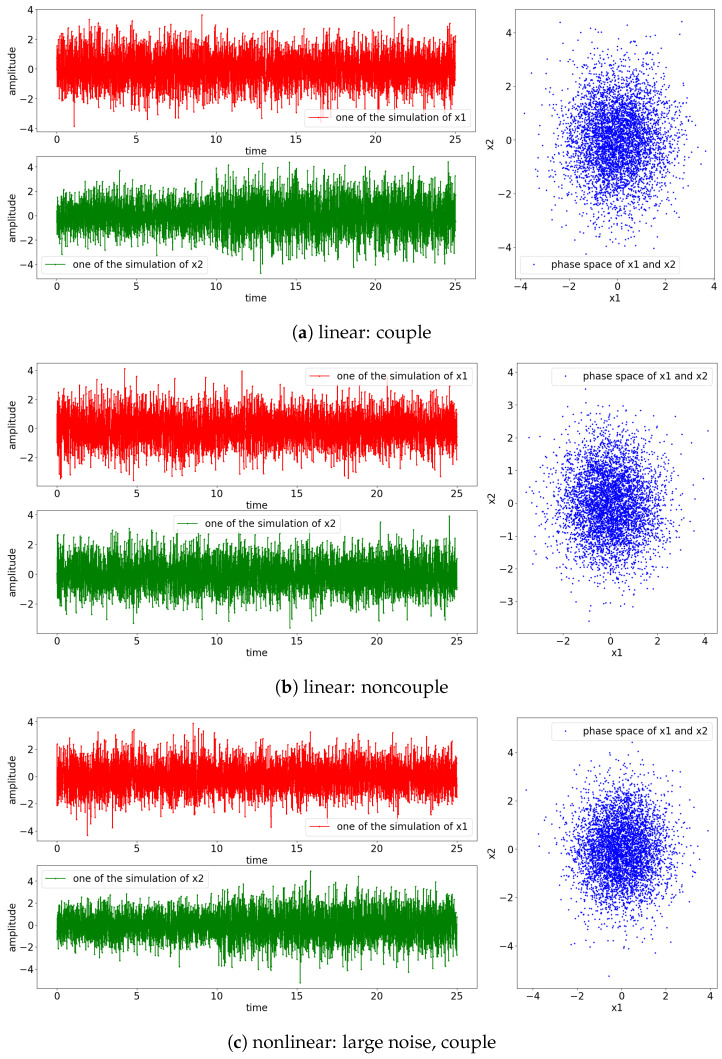
Result of simulation based on Equations (27) and (28) (refer to Figure 10 for graphical explanation of the process). The [top left, red] is the result of $x_{1}$ and [bottom, green] is the result of $x_{2}$; [right, blue] is the phase space of $x_{1}$ and $x_{2}$. Note that in the context of noncoupling, it is referring to H(τ−10)=0.

**Figure 12 entropy-25-00806-f012:**
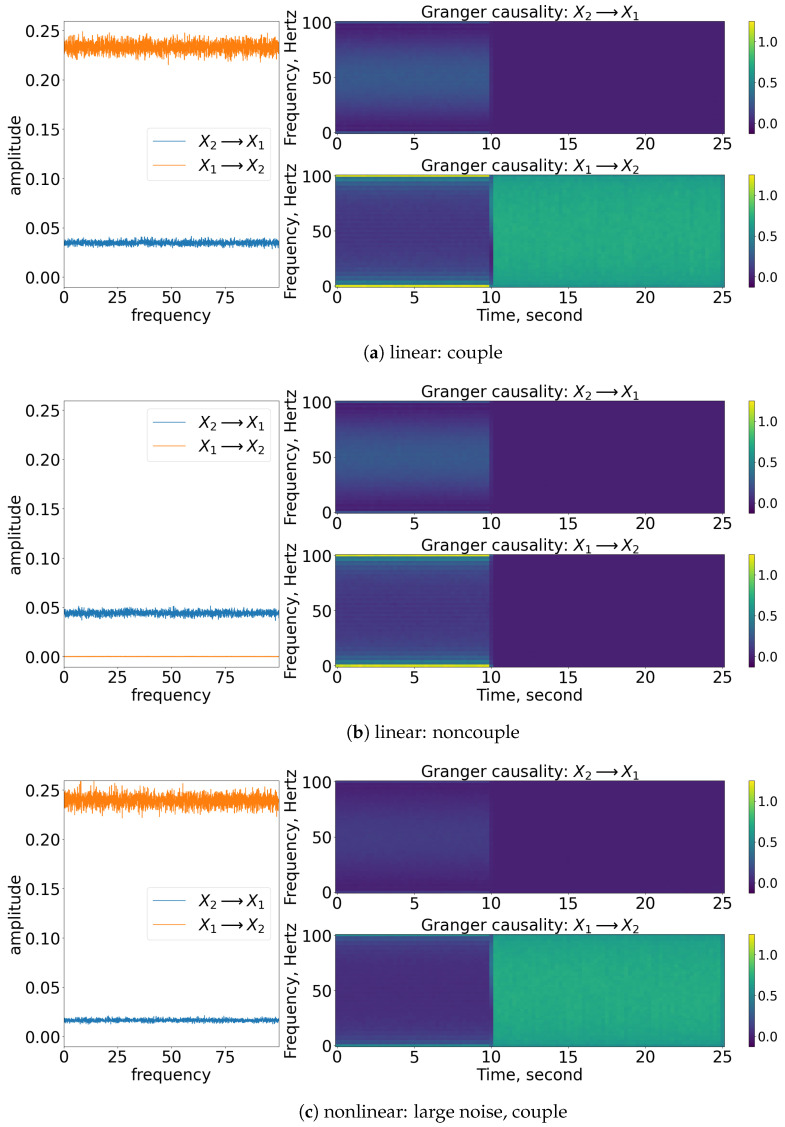
Result of the spectral and time-varying frequency of GC. Note that for each subfigure, the [left] shows the spectral (frequency domain) of GC, [blue] indicates $x_{2}$ causes $x_{1}$, while [orange] indicates $x_{1}$ causes $x_{2}$; [top, right] shows the time-varying frequency of GC of $x_{2}$ to $x_{1}$; [bottom, right] shows the time-varying frequency of GC of $x_{1}$ to $x_{2}$. The results shown are based on the processes in Equations (27) and (28). Refer to Figure 10 for the graphical explanation of the process. Note that the noncoupling is referring to H(τ−10)=0 for all the cases.

**Figure 13 entropy-25-00806-f013:**
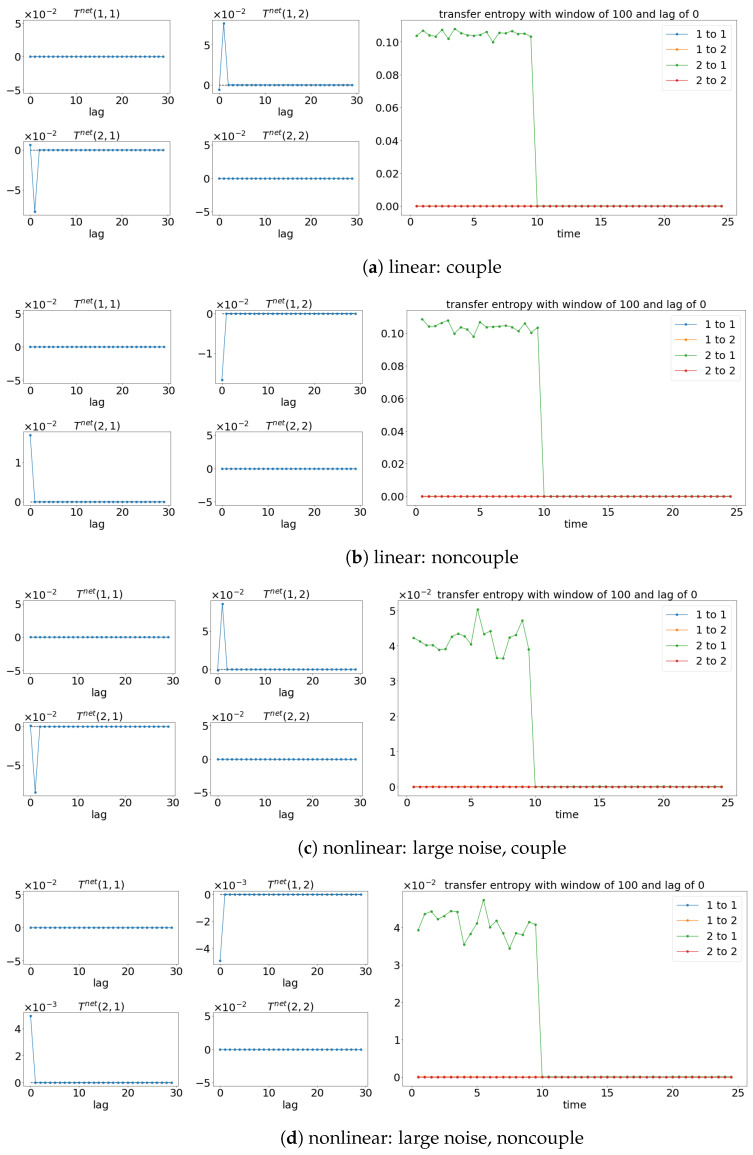
Result of the net TE and window sliding TE with the lag of 0. Note that for each subfigure, [left, 4×4] depicts the net TE for the whole signals, while the [right] shows the window sliding TE. The results shown are based on the processes in Equations (27) and (28). Refer to Figure 10 for the graphical explanation of the process. Note that the noncoupling is referring to H(τ−10)=0 for all the cases.

**Figure 14 entropy-25-00806-f014:**
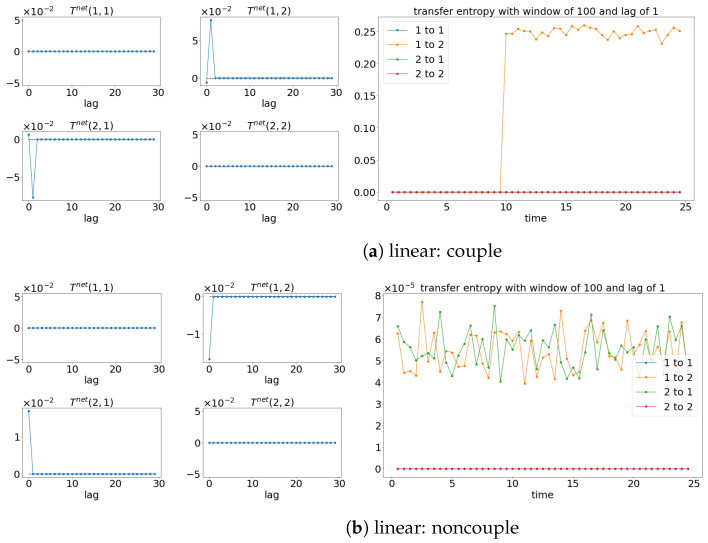
Result of the net TE and window sliding TE with the lag of 1. Note that for each subfigure, [left, 4×4] depicts the net TE for the whole signals, while the [right] shows the window sliding TE. The results shown are based on the processes in Equations (27) and (28). Refer to Figure 10 for the graphical explanation of the process. Note that the noncoupling is referring to H(τ−10)=0 for all the cases.

**Figure 15 entropy-25-00806-f015:**
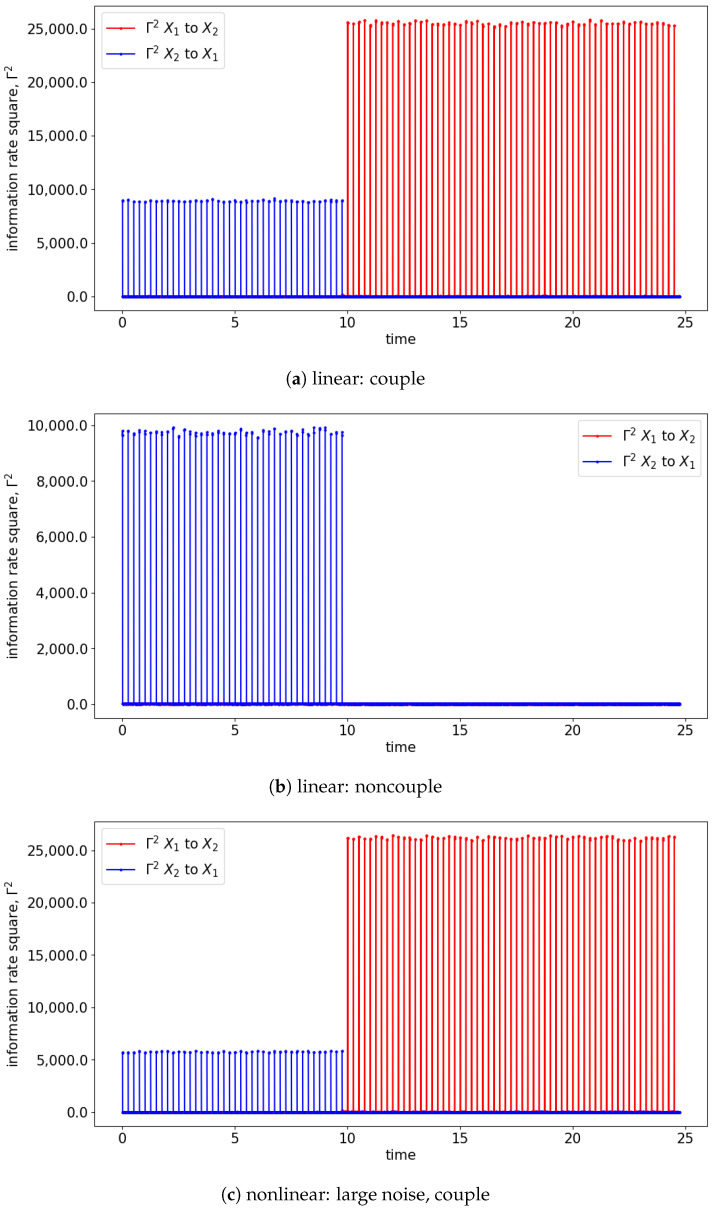
Result of information rate causality for Equations (27) and (28). Note that the interchange of the causality occurs at 10 s.

**Figure 16 entropy-25-00806-f016:**
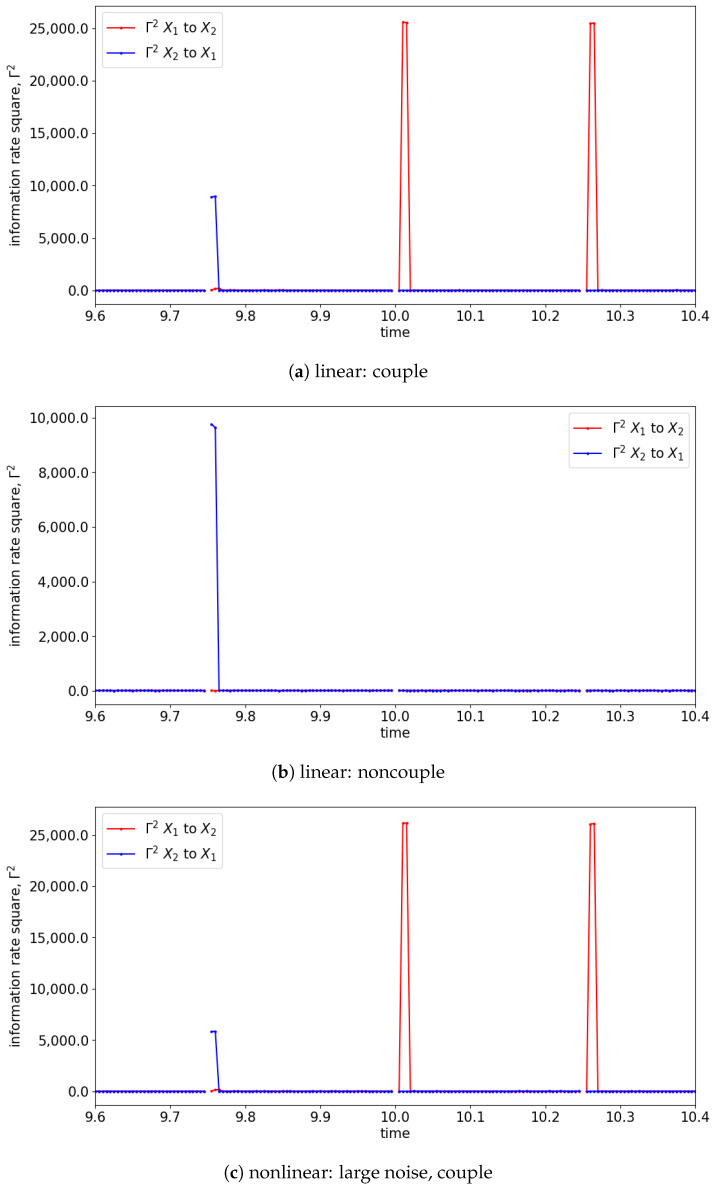
Zoom-in of Figure 15 for the information rate causality between 9.6 s and 10.4 s. Note that the interchange of the causality occurs at 10 s.

**Figure 17 entropy-25-00806-f017:**
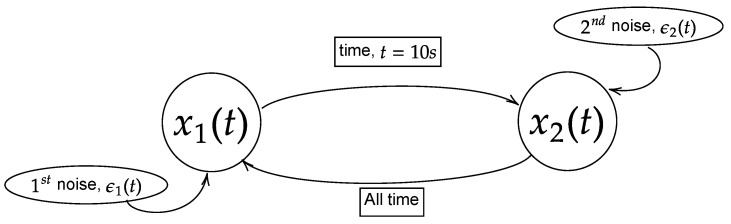
Model of the flow of information of the processes $x_{1}\left(t\right)$ and $x_{2}\left(t\right)$ for Equations (29) and (30). In this paper, the equations are simulated with physical time of 25 s and sampling frequency of 200 Hz (5000 realizations) with either large noise or small noise. Note that the $x_{2}\left(t\right)$ coupled with $x_{1}\left(t\right)$ throughout the process and the occurrence of bidirectional causality happens after 10 s. In the context of noncoupling, it is referring to H(τ−10)=0.

**Figure 18 entropy-25-00806-f018:**
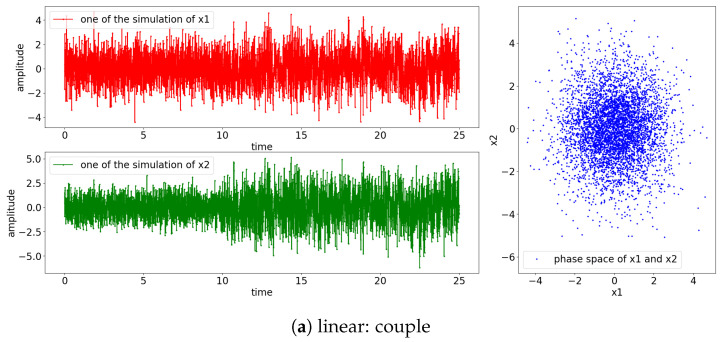
Result of simulation based on Equations (29) and (30) (refer to Figure 17 for graphical explanation of the process). The [top left, red] is the result of $x_{1}$; [bottom, green] shows the result of $x_{2}$; [right, blue] shows the phase space of $x_{1}$ and $x_{2}$. Note that the noncoupling in the context is referring to H(τ−10)=0.

**Figure 19 entropy-25-00806-f019:**
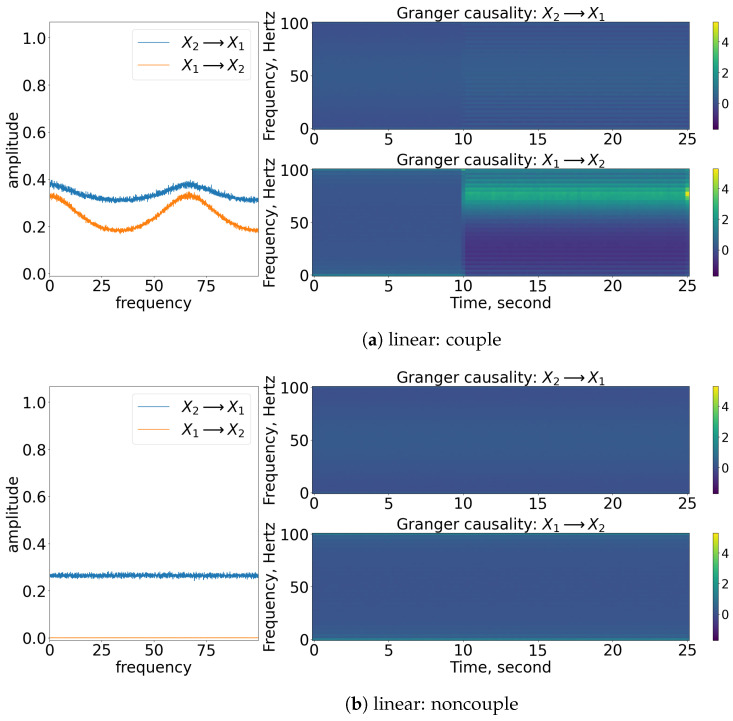
Result of the spectral and time-varying frequency of GC. Within each subfigure, the [left] shows the spectral (frequency domain) of GC, with [blue] showing that $x_{2}$ causes $x_{1}$ ($I_{x_{2}\to x_{1}}$) and [orange] indicating that $x_{1}$ causes $x_{2}$ ($I_{x_{1}\to x_{2}}$). The [top, right] figure shows that the time-varying frequency of GC of $x_{2}$ causes $x_{1}$ and [bottom, right] shows that $x_{1}$ causes $x_{2}$. The results shown are based on the expression in Equations (29) and (30). Refer to Figure 17 for the graphical explanation of the process. Note that the noncoupling is referring to H(τ−10)=0 for all the cases.

**Figure 20 entropy-25-00806-f020:**
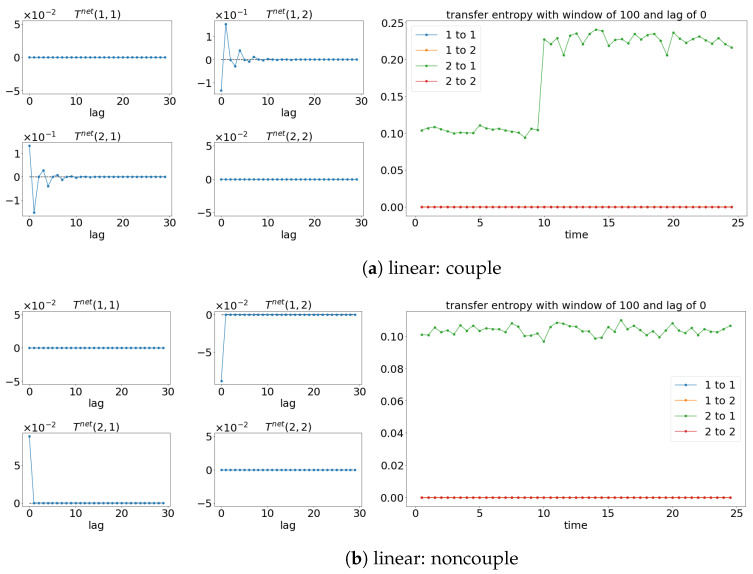
Result of the net TE and window sliding TE with the lag at 0. Within each subfigure, the [left, 4×4] figures show the net TE for the whole signals, while the [right] shows the window sliding TE. The results shown are based on the expression in Equations (29) and (30). Refer to Figure 17 for the graphical explanation of the process. Note that the noncoupling is referring to H(τ−10)=0 for all the cases.

**Figure 21 entropy-25-00806-f021:**
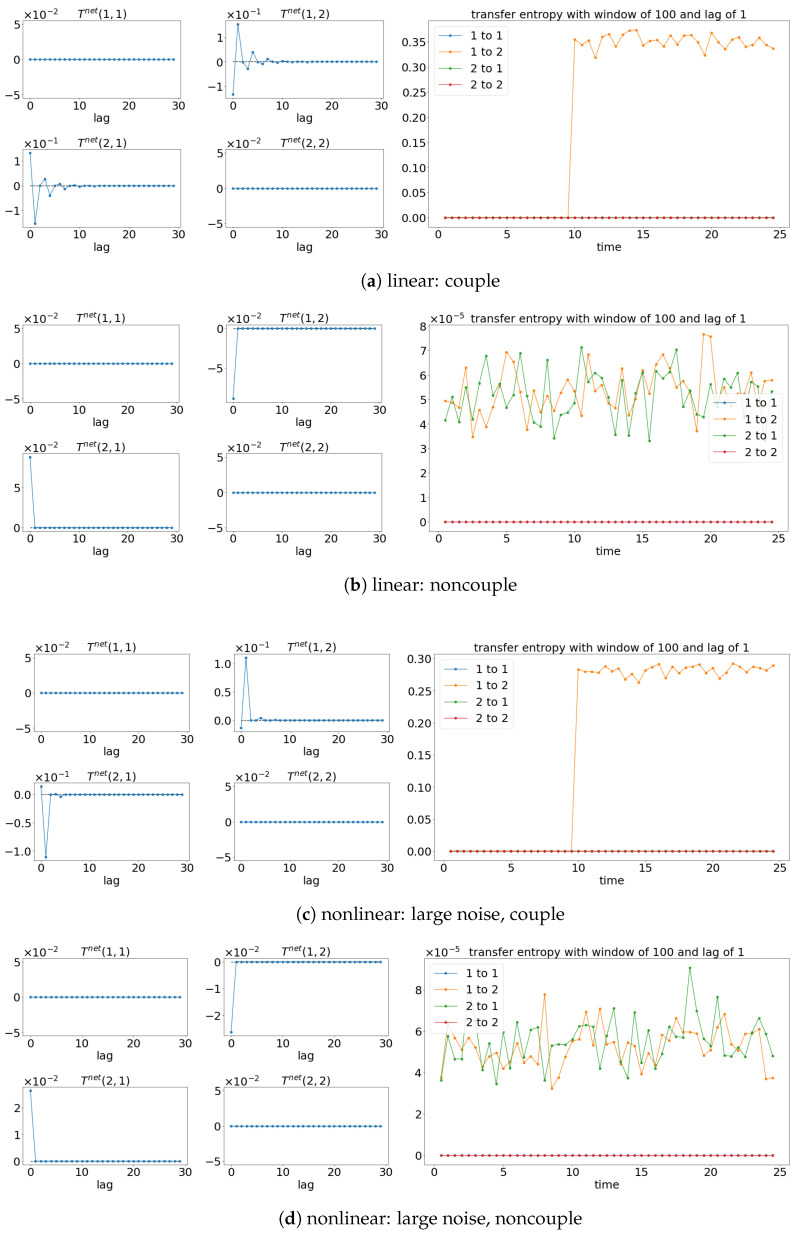
Result of the net TE and window sliding TE with the lag at 1. Within each subfigure, the [left, 4×4] figures show the net TE for the whole signals, while the [right] shows the window sliding TE. The results shown are based on the expression in Equations (29) and (30). Refer to Figure 17 for the graphical explanation of the process. Note that the noncoupling is referring to H(τ−10)=0 for all the cases.

**Figure 22 entropy-25-00806-f022:**
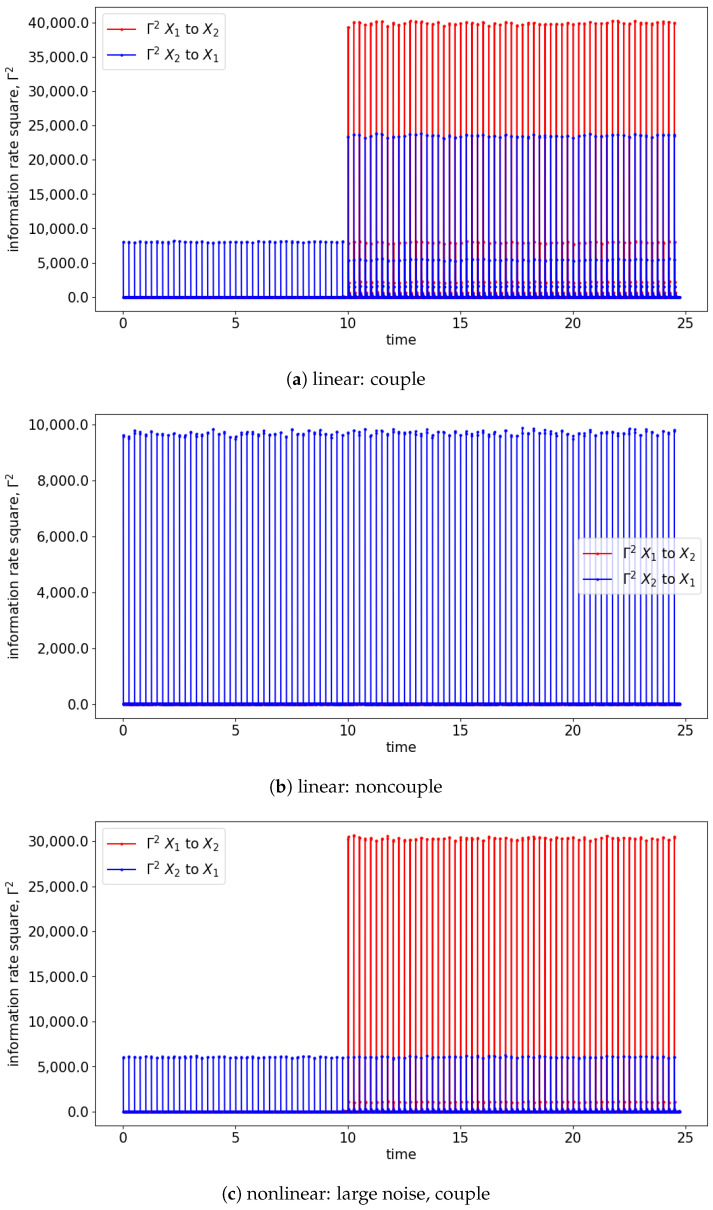
Result of information rate causality for Equations (29) and (30). Note that the bidirectional causation begins at 10 s. The results shown are based on the expression in Equations (29) and (30). Refer to Figure 17 for the graphical explanation of the process. Note that the noncoupling is referring to H(τ−10)=0 for all the cases.

**Figure 23 entropy-25-00806-f023:**
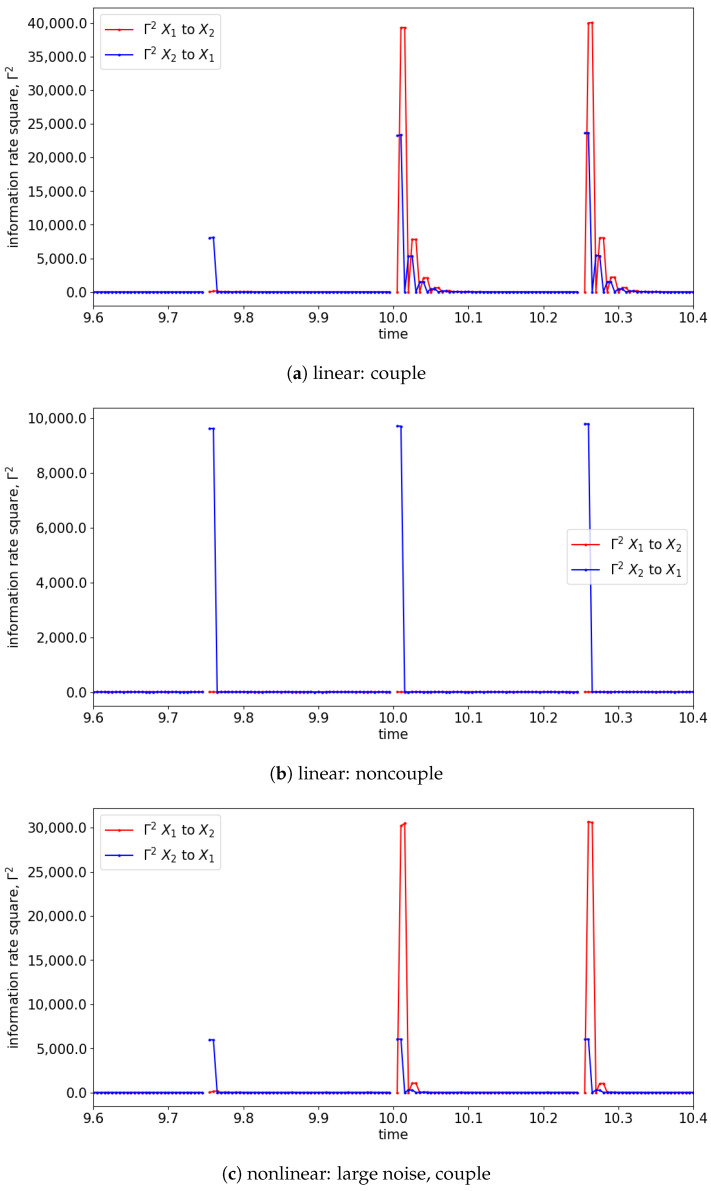
Zoom-in of Figure 22 for the information rate causality between 9.6 s and 10.4 s. Note that the bidirectional causation begins at 10 s. The results shown are based on the expression in Equations (29) and (30). Refer to Figure 17 for the graphical explanation of the process. Note that the noncoupling is referring to H(τ−10)=0 for all the cases.

**Table 1 entropy-25-00806-t001:** Summary of the performance of GC, TE, and information rate causality for different models discussed in Section 3, Section 4 and Section 5. In general, GC performs well for linear and stationary signals; TE performs well only if the lags are chosen correctly; information rate causality performs well and can recover the underlying lags of the signals.

Method	Unidirectional (Section 3)		Interchange (Section 4)		Bidirectional (Section 5)	
	Linear	Nonlinear	Linear	Nonlinear	Linear	Nonlinear
**GC**	performs well	fails in nonstationary signals	performs well		fails to capture the bidirectional coupling of signals	
**TE**	performs well if correct lags are chosen or else produces spurious results					
**Information rate causality**	performs well and able to recover the underlying lags of coupling					

## Data Availability

The data that support the findings of this study are available from the corresponding author upon reasonable request.

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
