# Peer review of "Causality Analysis with Information Geometry: A Comparison"

_entropy, 2023, doi:10.3390/e25050806_

Round 1

Reviewer 1 Report

The paper considers the problem of identifying and quantifying (Granger) causal effects between time varying processes. It evaluates the performance of three different approaches, based on VAR models, transfer entropy (TE) and the more recently introduced ‘information rate causality’ (IRC) metric, on compares them for various types of causal interactions, and shows that the latter outperforms the other two.

The problem is interesting, and very relevant in a large number of application areas. The problem becomes especially challenging when non-stationary systems or non-linear interactions are involved, and so any method or approach that can demonstrably improve on existing state-of-the-art is very welcome.

Unfortunately, the current contribution cannot fulfil that role: 

- main issue is that the paper essentially takes a few single case examples, and turns that into a basis for generic claims. That is simply not acceptable. 

- the main method itself (the IRC) is not new but already introduced in an earlier paper by one of the authors, which reduces the significance and novelty score of the current contribution,

- a minor gripe is that ‘Granger causality’ (a criterion) is taken synonymously with a straightforward implementation method (VAR), so that the conclusion is then ultimately that ‘information rate causality outperforms Granger causality’ … which is nonsensical as all three approaches (VAR, TE, and IRC) are simply different methods/metrics of deciding/quantifying whether one variable ‘Granger causes’ another

- in fact, there is not even a proper definition of ‘Granger causality’ anywhere in the current paper, which in turn implies there is no properly supported theoretical argument anywhere why one method would be more appropriate than another to decide on whether x1 is Granger causal to x2 or not, or under which condition/assumption,

- basically: VAR is simple but requires strong assumptions, TE is more flexible but typically requires more data, perhaps IRC is even more flexible (able to handle even weaker, more realistic assumptions) … but nowhere in the paper is there an attempt to properly assess the implications 

- the few models considered are extremely simple, containing only two processes x1 and x2. That is ok to illustrate some example behaviour or build some intuition, but in practice the real problem typically lies in applying e.g. TE to larger systems, involving more variables (curse of dimensionality). This aspect is currently completely ignored in the paper, which means we still have no idea how IRC will behave in any remotely realistic model, making the generic claims at the end somewhere between premature and pointless.

- the presentation of results for the three different approaches on the cases considered consists of a extensive sets of detailed results per instance … which in turn makes it almost impossible to gauge differences between results of the three approaches, let alone generic behaviour

- furthermore, in these types of problems it is necessary to distinguish between ‘presence’ of Granger causality between two variables (a yes/no type classification), and estimating / quantifying the strength. For the first a type of threshold is required, which can be varied to emphasise either precision or recall (or TPR, FPR etc.), which in turn implies the use of e.g. ROC curves / AUC to properly evaluate the difference in performance between two different approaches. None of this happens anywhere in the paper.

- similarly it would be extremely helpful to see dependence of the performance of each method relative to assumptions, size of available data, degree of nonlinearity etc.

- as a final remark I have to say the writing itself should be improved, as right now there are multiple sections where grammatical errors hinder understanding of the text, or lead to ambiguities on what is actually claimed or done.

In short: I am still excited by the potential of IRC as a method to assess causal relations in generic time-series domains, but for that I would need to see a much more thorough evaluation than provided in the current paper. Without it the contribution has little to offer over existing literature. Therefore at this stage I have to recommend ‘major revision’.

Reviewer 2 Report

In this manuscript, the authors propose a new method for detecting causal relationships among stochastic processes and compare it to two established methods, Granger causality and transfer entropy. The new method is applicable to non-stationary systems with nonlinear dynamics, like transfer entropy, but without the problem that transfer entropy has: the need to either compute a high-dimensional distribution or determine a single lag at which one variable maximally influences the dynamics of another.

However, many things are unclear in the manuscript that must be addressed before publication.

How is causality defined? Mainly I am asking whether the influence must be direct to be counted as causal, or if indirect influence is included in the definition.

The definition of the new method, “information rate causality”, in Eqns 20-21 depends on both t and t_i. What is t_i and how is it chosen? Is it integrated out? Is t or t_i plotted on the x-axis of the relevant figures in the text? How is this different from determining what lag to use in the transfer entropy? Figure 1 does not adequately explain this or the other methods.

Section 2.4 or the following sections should define the “uncoupled” cases. Which parameters are set to zero? What are the dynamical equations? This is hinted at in the figures where the input amplitude is plotted. Those plots take up a lot of space and contain little information. Their content should be explained in the text, instead.

In Section 2.4, data from many simulations are used (10,000 replicates), far more data than can typically be collected in an experiment. How relevant is the new method to analyzing experimental data? Are so many replicates needed for all of the methods?

Y-axes and color ranges of all panels of each figure should be the same. As is, it is difficult to compare the different cases because low-influence cases are plotted very zoomed in (take for example Fig 4 a,b).

Line 231 says that the information causality rate “quantifies the change of the probability distribution of a process caused by another”. This assumes the conclusion. It should be described in more elementary terms, eg “quantifies the rate of change of the probability distribution of one variable conditioned on another” or something equivalent.

Figure 7a, I would have said that there is no causal arrow from x2 to x1 in the linear coupled model. However, the information causality rate changes significantly once x1 begins to influence x2. This seems like the new method has failed. Can the authors explain this?

What is the difference between Fig 10 and Fig 8a? Why are there multiple peaks in both of them? Why does information causality rate oscillate?

Figure 9 should be a supplementary figure.

Throughout, the information causality length does not appear to add any new understanding. I recommend removing it from the paper completely.

The figure captions throughout need more explanation. Also the figure font sizes are too small to read.

“Net transfer entropy” is referred to in the introduction, before it is defined. I had never heard of this before. A brief in-line definition or description should be added to the intro.

Minor:

Error in line 426 “??”.

Reference 23 to Schreiber’s 2000 paper that defined transfer entropy is missing the journal and other details: it was published in Physical Review Letters.

Round 2

Reviewer 1 Report

I thank the authors for their response and clarifications. I can see they seriously tried to address the issues mentioned, and indeed the paper has become stronger as a result.

A number of concerns still stand, but I am willing to set these aside for the benefit of having the method itself out there. However two key issues remain (though I am confident these can be address adequately before publication):

1) The authors acknowledge in their response that the few specific examples used to demonstrate the method in the paper are not sufficient to draw general claims on performance, either on its own or relative to other methods. (e.g. 1st sentence of 2nd par. ’We are not claiming that our results are generic.’). However, the manuscript itself still reads very much like it does: e.g. final sentence of abstract: ‘Our results show that information rate causality can capture the coupling of both linear and nonlinear better than GC and TE.’).

As stated in the original review: I think the method is interesting and promising (which is why I am eager to see it go through), but currently it is largely based on a few, very specific examples that demonstrate working and potential, but that is not enough to suggest the strong claims on general performance or behaviour still present in the paper. 

2) I am concerned that the use of the Heaviside step function in the examples kicking in at t=10.0s (see e.g. eqs.24, 25) represents an extreme (nonlinear) discontinuity in the system, (resulting in the large peak around t=10 in fig.8a) that is not representative of the typical behaviour assumed in causally interacting processes. This particular case is very much tailored to the IRC rate of change metric, whereas other methods like VAR or TE typically assume much smoother, continuous (linear or nonlinear) interactions. As a result the latter two methods cannot be expected to perform well on the example cases provided, leading to a natural advantage for IRC that may not be indicative of the performance in standard cases. So while I do see the discontinuous case as an interesting and worthwhile example, I do think taking the continuous case (e.g. the eqs.24/25 without the step function) as the baseline reference for comparison would provide a more meaningful and relevant indication of the potential benefits of IRC relative to VAR and TE in practice. 

2b) As a final question related to this: could the authors provide some intuition for the peak at t=10.25s in fig8(a+c)? I understand the peak at t=10.0, but I do not understand from eq.24/25 why it would drop back to nearly zero in the interval [10.05,10.25] and then suddenly peak a second time at t=10.25. Is this a numerical artefact or does it truly represent another sudden shift in interaction, and if so, how can we recognise/understand this from eqs.24/25?

As stated: I am confident these issues can be addressed by the authors before final publication, and so I recommend 'minor revision'.
